# Locality Preserving Markovian Transition for Instance Retrieval

**Jifei Luo** [1]  **Wenzheng Wu** [1]  **Hantao Yao** [1]  **Lu Yu** [2]  **Changsheng Xu** [3]

## Abstract

Diffusion-based re-ranking methods are effective in modeling the data manifolds through similarity propagation in affinity graphs. However, positive signals tend to diminish over several steps away from the source, reducing discriminative power beyond local regions. To address this issue, we introduce the Locality Preserving Markovian Transition (LPMT) framework, which employs a long-term thermodynamic transition process with multiple states for accurate manifold distance measurement. The proposed LPMT first integrates diffusion processes across separate graphs using Bidirectional Collaborative Diffusion (BCD) to establish strong similarity relationships. Afterwards, Locality State Embedding (LSE) encodes each instance into a distribution for enhanced local consistency. These distributions are interconnected via the Thermodynamic Markovian Transition (TMT) process, enabling efficient global retrieval while maintaining local effectiveness. Experimental results across diverse tasks confirm the effectiveness of LPMT for instance retrieval.

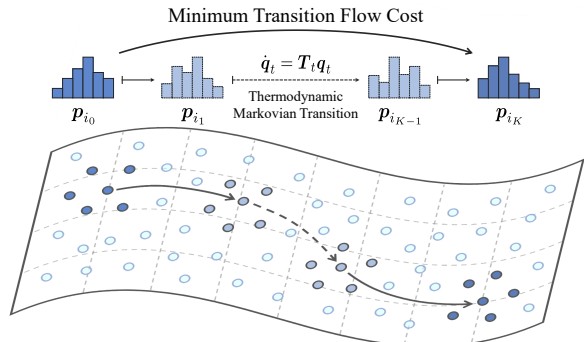

*Figure 1.* Illustration of Locality Preserving Markovian Transition. Each instance is embedded as a distribution within the manifold, with its characteristics shaped by the intrinsic local neighborhood structure. Distant distributions are bridged via multiple intermediate states, where each transition is confined to a local region and governed by the master equation. The minimum transition cost then serves as an effective distance measure for improved retrieval.

## 1. Introduction

Instance retrieval aims to identify images visually similar to a given query image on a large scale. Typically, the global image features are obtained through the aggregation of local descriptors (Jégou et al., 2012; Noh et al., 2017) or leveraging deep neural networks (Cao et al., 2020; Yang et al., 2021; Lee et al., 2022; 2023). After that, a ranking of images can be generated by computing the similarity or distance between these features. However, the feature extraction process inevitably loses some important information, and the capability of models often limits the expressiveness of the features. Consequently, refining the initial ranking results can improve the overall retrieval performance, referred to as re-ranking. A prominent approach is the Query Expansion (QE) (Chum et al., 2007; Shao et al., 2023), which uses high-confidence samples from the top-ranked results to generate a more robust query feature for secondary retrieval. However, these methods fail to effectively capture the latent manifold structure within the data space, limiting the performance.

Recently, diffusion-based methods (Iscen et al., 2017; 2018; Prokopchik et al., 2022; Bai et al., 2019a;c; Luo et al., 2024; Yang et al., 2019; Zhang et al., 2023) have been utilized to investigate the manifold structure of data for re-ranking, a process also known as manifold ranking. These methods begin with the initial retrieval results and construct a $k$-nearest neighbor graph (Zhou et al., 2003; Donoser & Bischof, 2013) to model the intrinsic data manifold. Once the graph is created, similarity information is iteratively propagated along the edges, allowing for the consideration of higher-order relationships between instances. The resulting manifold-aware similarity matrix demonstrates improved retrieval performance as the process converges. However, existing methods often rely heavily on graph construction strategies, *e.g.*, errors can propagate throughout the graph if the adjacency relationships are incorrect, while missing connections between high-confidence nodes can

[1]University of Science and Technology of China, Hefei, China [2]Tianjin University of Technology, Tianjin, China [3]Institute of Automation, Chinese Academy of Sciences, Beijing, China. Correspondence to: Hantao Yao <yaohantao@ustc.edu.cn>.

disrupt the flow of positive information. As a result, valuable information may diminish over multiple diffusion steps for instances outside the local region, leading to a loss of discriminative power. Therefore, improving the reliability of knowledge transmission in long-distance nodes is critical for effective manifold ranking.

The adverse effects of inaccurate propagation in diffusion-based methods can be substantially attenuated by modeling each instance as a probability distribution within the data manifold for distance measurement, while the importance of reliable neighborhoods can be emphasized at the same time. Additionally, building on the foundation of previous studies (Evans et al., 1993; Barato & Seifert, 2015; Ito, 2018; Van Vu & Saito, 2023), a long-term thermodynamic Markovian transition process consisting of multiple states can be utilized to quantify manifold distances. This approach effectively reduces information decay during the propagation across distributions. As demonstrated in Figure 1, two distant distributions are bridged through a series of intermediate states, each representing a probability distribution of an instance. By restricting each transition to a local region, the method ensures that information remains coherent and locally relevant throughout the process. The Markovian transition process governing each stage that connects two consecutive states is defined by the master equation (Seifert, 2012), which offers a more precise representation of the manifold structure than traditional metrics, such as total variation. This multi-state thermodynamic process creates a transition flow within the manifold, wherein the minimal cost effectively serves as a distance metric.

In this paper, we introduce a novel approach called Locality Preserving Markovian Transition (LPMT), which consists of Bidirectional Collaborative Diffusion (BCD), Locality State Embedding (LSE), and Thermodynamic Markovian Transition (TMT). The Bidirectional Collaborative Diffusion mechanism extends the reference adjacency graph into a graph set by systematically adding and removing some connections. This integration allows for a robust similarity matrix to be constructed through the joint optimization of combination weights and the equivalent objectives of the diffusion process, as highlighted in previous work (Luo et al., 2024). Subsequently, Locality State Embedding assigns a probability distribution to each instance within the manifold space, utilizing information from neighboring instances to enhance local consistency. Finally, the Thermodynamic Markovian Transition establishes a multi-state process on the manifold, wherein distant distributions navigate through several intermediate states within their respective regions. This approach elucidates the underlying manifold structure by capturing the minimum transition cost while preserving local characteristics. To compute the final distance, a weighted combination of this cost and the Euclidean distance is employed, enabling efficient global retrieval.

Experimental results on various instance retrieval tasks validate the effectiveness of the proposed Locality Preserving Markovian Transition. Specifically, LPMT achieves mAP scores of 84.7%/67.8% on $R$Oxf and 93.0%/84.1% on $R$Par under medium and hard protocols, respectively, demonstrating its superior performance.

## 2. Related Work

**Instance Retrieval**. The objective of instance retrieval is to identify images in a database that resemble the content of a query instance. With the rapid advancement of deep learning, global features extracted by deep neural networks (Radenović et al., 2019; Cao et al., 2020; Yang et al., 2021; Lee et al., 2022; 2023) have gradually replaced local descriptors (Lowe, 2004; Jégou et al., 2012; Noh et al., 2017). Despite their effectiveness, the retrieval performance can be further refined through a post-process known as re-ranking.

**Re-ranking**. Specifically, re-ranking can be broadly divided into Query Expansion, Diffusion-based Methods, Context-based Methods, and Learning-based Methods.

*Query expansion*. The higher relevance maintained by the top-ranked images leads to the development of Query Expansion (QE), which integrates neighboring features to build a more effective query. While AQE (Chum et al., 2007) simply averages the features of top returned images, AQEwD (Gordo et al., 2017), DQE (Arandjelović & Zisserman, 2012), $\alpha$QE (Radenović et al., 2019), and SG (Shao et al., 2023) apply diminishing aggregation weights to the subsequent ones, leading to enhanced retrieval performance.

*Diffusion-based Methods*. Leveraging the intrinsic manifold structure of data, diffusion-based methods serve as a powerful technique for re-ranking. After the theory has originally been developed (Zhou et al., 2003; Yang et al., 2009; Donoser & Bischof, 2013), it has been successfully introduced to the field of instance retrieval (Iscen et al., 2017; 2018; Yang et al., 2019). To further capture the underlying relationships, researchers (Zhou et al., 2012; Bai et al., 2019a;c; Yang et al., 2013; Zhang et al., 2015) seek to aggregate higher-order information by propagating messages on a hypergraph or integrating information from distinct graphs. Additionally, EGT (Chang et al., 2019) and CAS (Luo et al., 2024) adjust the diffusion strategy to address the problem of unreliable connections, resulting in improved effectiveness.

*Context-based Methods*. Given that the contextual information contained by nearest neighbors can lead to notable improvements in retrieval performance. Pioneer works (Jégou et al., 2007; Shen et al., 2012; Sarfraz et al., 2018) adjust the distance measure by using the ranking or similarity relationships in the neighborhood, while recent approaches (Bai & Bai, 2016; Zhong et al., 2017; Zhang et al., 2020; Yu et al., 2023; Liao et al., 2023; Kim et al., 2022) encode each

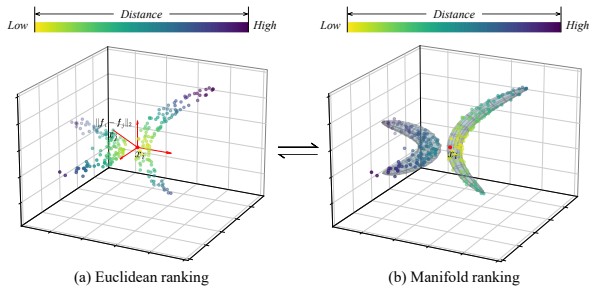

*Figure 2.* Comparison of ranking results based on (a) Euclidean distance and (b) manifold-aware distance in the feature space.

instance into a manifold-aware space to perform re-ranking, where similar images exhibit higher contextual consistency.

*Learning-based Methods.* Recently, deep learning methods have also been introduced to assist with re-ranking. For example, Gordo et al. (2020) and Ouyang et al. (2021) leverage the robust encoding power of self-attention mechanisms to learn weight relationships for aggregating representative descriptors. Meanwhile, Liu et al. (2019) and Shen et al. (2021) seek to perform information propagation via optimizing graph neural network (Gasteiger et al., 2018), allowing the ranking result to capture the intrinsic manifold structure.

**Large Pre-trained Models for Information Retrieval**. Information retrieval in both visual and textual modalities has undergone a fundamental shift with the emergence of large pre-trained models, often referred to as foundation models.

*VLMs for Image Retrieval.* Trained on colossal web-scale image-text datasets, Vision-Language Models (VLMs) (Radford et al., 2021; Jia et al., 2021; Yao et al., 2022; Li et al., 2022; 2023; Liu et al., 2023) serve as foundational backbones for diverse vision and language tasks. These models provide semantically rich and discriminative features for both modalities, aligned within a shared semantic space, enabling sophisticated cross-modal retrieval. However, unlike precise instance-level retrieval, VLMs inherently prioritize broad conceptual understanding, often yielding initial results that are semantically relevant but lack fine-grained instance discrimination (*e.g.*, a query for "Eiffel Tower" may retrieve a range of related images without accurately distinguishing between highly similar photographic instances). Nevertheless, the latent manifold structure present in the feature space of semantically similar samples offers a compelling basis for subsequent re-ranking methods aimed at refining retrieval precision.

*LLMs for Textual Retrieval.* Moving beyond lexical statistics, Large Language Models (LLMs) (Thakur et al., 2021; Devlin et al., 2019) empower deep semantic understanding of both queries and documents. Through the encoding of queries and documents into a high-dimensional embedding space, LLMs are broadly applied within dense information retrieval paradigms. Building on these retrieval capabilities,

LLMs are further integrated into Retrieval-Augmented Generation (RAG) systems (Chen et al., 2024; Borgeaud et al., 2022; Asai et al., 2024), enabling the extraction of relevant content from external knowledge bases to produce coherent and factually accurate responses. Given the modern emphasis on logical reasoning and deeper understanding in textual retrieval (Su et al., 2025), leveraging powerful LLMs (Xiao et al., 2023) to dynamically re-evaluate query-document similarity for re-ranking yields superior quality, surpassing approaches solely based on semantic feature embeddings.

## 3. Preliminary

Given a query image, instance retrieval aims to sort the gallery image set in ascending order, where images at the front are more similar to the query. Formally, define the whole image set containing query and gallery images as $\mathcal{X} = \{x_1, x_2, \ldots, x_n\}$. A $d$-dimensional image feature for each image in $\mathcal{X}$ can be extracted with a pretrained model to measure the pairwise distance. Denote the image feature corresponds to $x_i$ as $\boldsymbol{f}_i$, the Euclidean distance between $x_i$ and $x_j$ in the feature space can be calculated by:

$$d(i, j) = \|\boldsymbol{f}_i - \boldsymbol{f}_j\|_2. \tag{1}$$

The distance between the query $x_i$ and images in $\mathcal{X}$ computed by Eq. (1) can be directly used for ranking the gallery set concerning $x_i$. However, the retrieval result based on Euclidean distance always reflects a sub-optimal performance.

Drawing from the prior knowledge that similar images are distributed along a low-dimensional manifold induced by the whole image set $\mathcal{X}$. Prioritizing images that simultaneously exhibit higher proximity in Euclidean and manifold spaces can improve retrieval results. To achieve this, a general approach is to model the underlying manifold structure with a $k$-nearest neighbor graph $\mathcal{G} = \{\mathcal{V}, \mathcal{E}\}$, each vertex $v_i$ in $\mathcal{V} = \{v_1, v_2, \ldots, v_n\}$ represents the corresponding position of $x_i$ within the data manifold, while $\mathcal{E} = \mathcal{V} \times \mathcal{V}$ denotes the edges weighted by:

$$\boldsymbol{W}_{ij} = \mathbb{1}_{ij}^{\mathcal{N}} \exp\left(-d^2(i, j)/\sigma^2\right), \tag{2}$$

where $\mathbb{1}^{\mathcal{N}}$ is an indicator matrix to represent the $k$-nearest neighbors, *i.e.*, $\mathbb{1}_{ij}^{\mathcal{N}} = 1$, if $j \in \mathcal{N}(i, k)$, otherwise $\mathbb{1}_{ij}^{\mathcal{N}} = 0$.

To exploit the underlying manifold geometry encoded in the adjacency matrix $\boldsymbol{W}$, diffusion-based methods (Iscen et al., 2017; Bai et al., 2019a; Luo et al., 2024) spread information along neighboring edges within the graph in an unsupervised manner, producing a similarity matrix that captures the manifold relationships for re-ranking. A comprehensive visual explanation of manifold ranking is shown in Fig. 2. Nevertheless, the diffusion process is susceptible to interference from inaccuracies in the adjacency graph construction and struggles to maintain discriminative power beyond the local region, thereby limiting the overall performance.

# 4. Locality Preserving Markovian Transition

To overcome the limitations that traditional diffusion models are sensitive to graph construction and struggle to generalize across varying data distributions, we propose Bidirectional Collaborative Diffusion (BCD) to integrate diffusion processes on multi-level affinity graphs automatically. To further enhance global discriminative power without compromising local effectiveness, we introduce the Locality State Embedding (LSE) strategy, which represents each instance as a local consistent distribution within the underlying manifold. Afterwards, Thermodynamic Markovian Transition (TMT) is proposed to perform a constrained time evolution transition process within the local regions at each stage. The minimal cost of multi-step transitions can effectively capture the intrinsic manifold structure, providing a powerful distance metric for instance retrieval. In the following, we give a detailed description of each proposed component.

## 4.1. Bidirectional Collaborative Diffusion

Given a reference adjacency matrix $\boldsymbol{W}$, we can generate an extended set $\{\boldsymbol{W}^1, \boldsymbol{W}^2, \ldots, \boldsymbol{W}^m\}$ by adding and removing some connections. Their contribution to the total diffusion process is denoted by $\{\beta_v\}_{v=1}^m = \{\beta_1, \beta_2, \ldots, \beta_m\}$, which can be dynamically adjusted. As shown in Fig. 3, BCD seeks to automatically perform diffusion and integration to produce a robust result. The smoothed similarity $\boldsymbol{F}$ and weights $\beta$ are jointly optimized follow by:

$$\min_{\beta,\boldsymbol{F}} \quad \sum_{v=1}^m \beta_v H^v + \frac{1}{2}\lambda\|\beta\|_2^2$$
$$\text{s.t.} \quad 0 \le \beta_v \le 1, \sum_{v=1}^m \beta_v = 1, \tag{3}$$

where $H^v$ denotes the objective value of Bidirectional Similarity Diffusion (Bai et al., 2019a;c; Luo et al., 2024) process with respect to $\boldsymbol{W}^v$, followed as:

$$H^v = \frac{1}{4}\sum_{k=1}^n \sum_{i,j=1}^n \boldsymbol{W}_{ij}^v \left(\frac{\boldsymbol{F}_{ki}}{\sqrt{\boldsymbol{D}_{ii}^v}} - \frac{\boldsymbol{F}_{kj}}{\sqrt{\boldsymbol{D}_{jj}^v}}\right)^2$$
$$+ \boldsymbol{W}_{ij}^v \left(\frac{\boldsymbol{F}_{ik}}{\sqrt{\boldsymbol{D}_{ii}^v}} - \frac{\boldsymbol{F}_{jk}}{\sqrt{\boldsymbol{D}_{jj}^v}}\right)^2 + \mu\|\boldsymbol{F} - \boldsymbol{E}\|_F^2. \tag{4}$$

As for each adjacency matrix $\boldsymbol{W}^v$ in Eq. (4), $\boldsymbol{D}^v$ is the diagonal matrix with its $i$-th diagonal element equal to the summation of the $i$-th row in $\boldsymbol{W}^v$. The regularization term is weighted by $\mu > 0$, where the matrix $\boldsymbol{E}$ is positive and semi-definite and is used to avoid $\boldsymbol{F}$ from being extremely smooth. For a given triplet of vertices $v_k$, $v_i$ and $v_j$ in graph $\mathcal{G}$, the smoothed similarities $\boldsymbol{F}_{ki}$ and $\boldsymbol{F}_{kj}$ are regularized by the affinity weight $\boldsymbol{W}_{ij}$. Meanwhile, the bidirectional strategy also takes the reverse pair $\boldsymbol{F}_{ik}$ and $\boldsymbol{F}_{jk}$ into consideration, ensuring the symmetric of the target matrix $\boldsymbol{F}$.

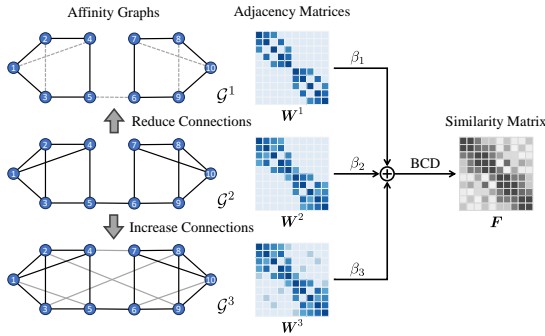

*Figure 3.* An illustration of the proposed Bidirectional Collaborative Diffusion (BCD) algorithm with three scales, where different connection strategies are employed to accommodate diverse data distributions. BCD automatically performs diffusion and integration to generate a robust similarity matrix.

The optimization problem in Eq. (3), which simultaneously depends on $\beta$ and $\boldsymbol{F}$, is inherently complex and impractical for a direct solution. To resolve this, we propose a numerical method that decomposes the target function into two sub-problems: *Optimize $\boldsymbol{F}$ with Fixed $\beta$* and *Optimize $\beta$ with Fixed $\boldsymbol{F}$*. This allows for a systematic iterative approximation of the optimal result through the fixed-point scheme.

*Optimize $\boldsymbol{F}$ with Fixed $\beta$.* Assuming that $\{\beta_v\}_{v=1}^m$ are fixed, such that they can be directly omitted during the optimization process of $\boldsymbol{F}$. Incorporating the definition of $H^v$ into the objective yields the following formulation:

$$\min_{\boldsymbol{F}} \frac{1}{4}\sum_{v=1}^m \sum_{k=1}^n \sum_{i,j=1}^n \beta_v \boldsymbol{W}_{ij}^v \left(\frac{\boldsymbol{F}_{ki}}{\sqrt{\boldsymbol{D}_{ii}^v}} - \frac{\boldsymbol{F}_{kj}}{\sqrt{\boldsymbol{D}_{jj}^v}}\right)^2$$
$$+ \beta_v \boldsymbol{W}_{ij}^v \left(\frac{\boldsymbol{F}_{ik}}{\sqrt{\boldsymbol{D}_{ii}^v}} - \frac{\boldsymbol{F}_{jk}}{\sqrt{\boldsymbol{D}_{jj}^v}}\right)^2 + \mu\|\boldsymbol{F} - \boldsymbol{E}\|_F^2. \tag{5}$$

This formulation is still hard to deal with, refer to the derivation in Appendix A.1, we transform it into a vectorized formulation to facilitate the solution as follows,

$$J = \sum_{v=1}^m \beta_v vec(\boldsymbol{F})^\top (\mathbb{I} - \bar{\mathbb{S}}^v) vec(\boldsymbol{F}) + \mu\|vec(\boldsymbol{F} - \boldsymbol{E})\|_2^2, \tag{6}$$

where $\mathbb{I} \in \mathbb{R}^{n^2 \times n^2}$ is an identity matrix, and $\bar{\mathbb{S}}^v$ is the mean Kronecker product form of normalized matrix $\boldsymbol{S}^v$, calculated by $\bar{\mathbb{S}}^v = (\boldsymbol{I} \otimes \boldsymbol{S}^v + \boldsymbol{S}^v \otimes \boldsymbol{I})/2$, in which the corresponding normalized matrix to $\boldsymbol{W}^v$ is denoted as $\boldsymbol{S}^v = (\boldsymbol{D}^v)^{-1/2}\boldsymbol{W}^v(\boldsymbol{D}^v)^{-1/2}$. Additionally, $vec(\cdot)$ denotes the vectorization operator, with its inverse function as $vec^{-1}(\cdot)$.

The Hessian matrix of $J$ in Eq. (6) is proved to be positive-definite in Appendix A.1, such that the optimal solution is achieved when the first order is equal to zero, that is:

$$\nabla_{vec(\boldsymbol{F})} J = \sum_{v=1}^m \beta_v(2\mathbb{I} - \bar{\mathbb{S}}^v) vec(\boldsymbol{F}) + 2\mu(vec(\boldsymbol{F} - \boldsymbol{E})). \tag{7}$$

By substituting the hyper-parameter $\beta_v$ with $\alpha_v = \frac{\beta_v}{\mu+1}$ and introduce $\alpha = \frac{1}{\mu+1}$, the simplified closed form solution $\boldsymbol{F}^*$ can be expressed as:

$$\boldsymbol{F}^* = (1-\alpha)vec^{-1}\big((\mathbb{I} - \sum_{v=1}^m \alpha_v \bar{\mathbb{S}}^v)^{-1}vec(\boldsymbol{E})\big). \quad (8)$$

Directly solving the inverse of a matrix with $n^2 \times n^2$ dimensions is still computationally infeasible. To address this problem, we adopt the iterative approach to gradually approach the optimal solution, followed by[1]:

$$\boldsymbol{F}^{(t+1)} = \frac{1}{2}\sum_{v=1}^m \alpha_v \big(\boldsymbol{F}^{(t)}(\boldsymbol{S}^v)^\top + \boldsymbol{S}^v\boldsymbol{F}^{(t)}\big) + (1-\alpha)\boldsymbol{E}. \quad (9)$$

Inspired by Iscen et al. (2017), the rate of convergence can be further enhanced using the conjugate gradient method, requiring fewer iterations as shown in Algorithm 2.

*Optimize $\beta$ with Fixed $\boldsymbol{F}$.* In the situation when $\boldsymbol{F}$ is fixed, we can directly compute the objective value for each $H^v$ with Eq. (4), and we only need to optimize $\beta$ by solving:

$$\min_\beta \quad \sum_{v=1}^m \beta_v H^v + \frac{1}{2}\lambda\|\beta\|_2^2$$
$$\text{s.t.} \quad 0 \le \beta_v \le 1, \sum_{v=1}^m \beta_v = 1. \quad (10)$$

Deriving an analytical solution for this Lasso-form optimization problem is still challenging due to inequality constraints. Therefore, a general approach involves iteratively updating each $\beta_v$ through coordinate descent. However, in this case, we propose a more efficient strategy that allows updating all $\beta$ values simultaneously, explicitly eliminating the need for iteration. As demonstrated in Appendix A.2, we can establish an equivalent condition that identifies the valid index set where the inequality constraints are slack, as follows:

$$\mathcal{I} = \{v|H^v < (\sum_{v'\in\mathcal{I}} H^{v'} + \lambda)/|\mathcal{I}|, v = 1, 2, \ldots, m\}, \quad (11)$$

while other $\{\beta_v\}_{v\notin\mathcal{I}}$ can be proved to be 0. All the $\{\beta_v\}_{v\in\mathcal{I}}$ will not violate the inequality constraint $0 \le \beta_v \le 1$, such that it can be ignored for a simpler formulation. By introducing a Lagrangian multiplier $\eta$, the Lagrangian function $\mathcal{L}(\beta, \eta)$ corresponds to the primal optimization problem in Eq. (10) without inequality constraints is formulated as:

$$\mathcal{L}(\beta, \eta) = \sum_{v\in\mathcal{I}} \beta_v H^v + \frac{1}{2}\lambda\|\beta\|_2^2 + \eta(1 - \sum_{v\in\mathcal{I}} \beta_v). \quad (12)$$

The optimal result can then be obtained by solving the KKT conditions, with the optimal weight set $\{\beta_v^*\}_{v=1}^m$ given by:

$$\beta_v^* = \begin{cases} \dfrac{\sum_{v'\in\mathcal{I}} H^{v'} - |\mathcal{I}|H^v + \lambda}{\lambda|\mathcal{I}|}, v \in \mathcal{I}, \\ 0, \quad v \in \{1, 2, \ldots, m\}/\mathcal{I}. \end{cases} \quad (13)$$

---

[1]For a comprehensive derivation, refer to Appendix A.

---

**Algorithm 1** Bidirectional Collaborative Diffusion

**Input:** extended adjacency matrices set $\{\boldsymbol{W}^v\}_{v=1}^m$, hyper-parameters $\lambda$, $\mu$, max number of iterations $maxiter$.
**Output:** adaptive smoothed similarity matrix $\boldsymbol{F}$.
1: initialize $t = 0$, $\{\beta_v\}_{v=1}^m = 1/m$ and $\boldsymbol{F}^{(0)} = \boldsymbol{E}$
2: **repeat**
3:     update $\boldsymbol{F}$ with fixed $\beta$ following Eq. (9)
4:     compute $\{H^v\}_{v=1}^m$ following Eq. (4)
5:     set $\{\beta_v\}_{v=1}^m$ with 0
6:     filter the valid index set $\mathcal{I}$ following Eq. (11)
7:     update $\{\beta_v\}_{v\in\mathcal{I}}$ with fixed $\boldsymbol{F}$ following Eq. (13)
8:     $t \leftarrow t + 1$
9: **until** convergence or $t = maxiter$

---

*Overall Optimization.* As shown in the Algorithm 1, the overall optimization problem for Bidirectional Collaborative Diffusion can be solved by recursively optimizing $\boldsymbol{F}$ and $\beta$ until convergence. The resulting $\boldsymbol{F}^*$ can effectively capture the underlying manifold structure and is less sensitive to the construction strategy of the affinity graph.

### 4.2. Locality State Embedding

The obtained smooth similarity matrix $\boldsymbol{F}^*$ demonstrates significant improvements in accurately capturing neighborhood relationships, especially in local regions. To preserve local reliability and facilitate further exploration of potential manifold structure information, the proposed Locality State Embedding (LSE) leverages this similarity matrix to map each instance into the manifold space with an $n$-dimensional distribution, where each dimension is coupled with a node in graph $\mathcal{G}$. To effectively mitigate the negative impact of noise within neighborhoods, we employ the $k$-reciprocal strategy to determine local regions for weight assignment, which can be formally expressed as:

$$\mathcal{R}(i, k) = \{j|(j \in \mathcal{N}(i, k)) \wedge (i \in \mathcal{N}(j, k))\}. \quad (14)$$

To avoid ambiguity, we use $k_1$ to represent the size of the local region. For each instance, only the indices belonging to the local region are preserved and assigned the weights by utilizing the corresponding row of $\boldsymbol{F}^*$ followed by an $l_1$ regularization, resulting in a sparse matrix $\hat{\boldsymbol{P}}$ followed by:

$$\hat{\boldsymbol{P}}_{ij} = \boldsymbol{F}_{ij}^* / \sum_{j\in\mathcal{R}(i,k_1)} \boldsymbol{F}_{ij}^*, \text{ if } j \in \mathcal{R}(i, k_1), \quad (15)$$

where $\hat{\boldsymbol{P}} = [\hat{\boldsymbol{p}}_1, \ldots, \hat{\boldsymbol{p}}_n]^\top$, and each $\hat{\boldsymbol{p}}_i$ represents a state distribution for $x_i$. In a sense, alongside the BCD process described in Section 4.1, the mapping process driven by LSE serves as a kernel function that transforms a $d$-dimensional feature into an $n$-dimensional manifold-aware state distribution. Given that the close neighbors have high confidence in belonging to the same category, such that the probability distributions within the local neighborhood $\mathcal{N}(i, k_2)$ ($k_2 < k_1$)

*Table 1.* Evaluation of the performance on $R$Oxf, $R$Par, $R$Oxf+1M, $R$Par+1M. Using R-GeM (Radenović et al., 2019) as the baseline.

| Method | Medium | | | | Hard | | | |
|---|---|---|---|---|---|---|---|---|
| | $R$**Oxf** | $R$**Oxf+1M** | $R$**Par** | $R$**Par+1M** | $R$**Oxf** | $R$**Oxf+1M** | $R$**Par** | $R$**Par+1M** |
| R-GeM (Radenović et al., 2019) | 67.3 | 49.5 | 80.6 | 57.4 | 44.2 | 25.7 | 61.5 | 29.8 |
| AQE (Chum et al., 2007) | 72.3 | 56.7 | 82.7 | 61.7 | 48.9 | 30.0 | 65.0 | 35.9 |
| $\alpha$QE (Radenović et al., 2019) | 69.7 | 53.1 | 86.5 | 65.3 | 44.8 | 26.5 | 71.0 | 40.2 |
| DQE (Arandjelović & Zisserman, 2012) | 70.3 | 56.7 | 85.9 | 66.9 | 45.9 | 30.8 | 69.9 | 43.2 |
| AQEwD (Gordo et al., 2017) | 72.2 | 56.6 | 83.2 | 62.5 | 48.8 | 29.8 | 65.8 | 36.6 |
| LAttQE (Gordo et al., 2020) | 73.4 | 58.3 | 86.3 | 67.3 | 49.6 | 31.0 | 70.6 | 42.4 |
| ADBA+AQE | 72.9 | 52.4 | 84.3 | 59.6 | 53.5 | 25.9 | 68.1 | 30.4 |
| $\alpha$DBA+$\alpha$QE | 71.2 | 55.1 | 87.5 | 68.4 | 50.4 | 31.7 | 73.7 | 45.9 |
| DDBA+DQE | 69.2 | 52.6 | 85.4 | 66.6 | 50.2 | 29.2 | 70.1 | 42.4 |
| ADBAwD+AQEwD | 74.1 | 56.2 | 84.5 | 61.5 | 54.5 | 31.1 | 68.6 | 33.7 |
| LAttDBA+LAttQE | 74.0 | 60.0 | 87.8 | 70.5 | 54.1 | 36.3 | 74.1 | 48.3 |
| DFS (Iscen et al., 2017) | 72.9 | 59.4 | 89.7 | 74.0 | 50.1 | 34.9 | 80.4 | 56.9 |
| RDP (Bai et al., 2019a) | 75.2 | 55.0 | 89.7 | 70.0 | 58.8 | 33.9 | 77.9 | 48.0 |
| EIR (Yang et al., 2019) | 74.9 | 61.6 | 89.7 | 73.7 | 52.1 | 36.9 | 79.8 | 56.1 |
| EGT (Chang et al., 2019) | 74.7 | 60.1 | 87.9 | 72.6 | 51.1 | 36.2 | 76.6 | 51.3 |
| CAS (Luo et al., 2024) | 80.7 | 61.6 | 91.0 | 75.5 | 64.8 | 39.1 | 80.7 | 59.7 |
| GSS (Liu et al., 2019) | 78.0 | 61.5 | 88.9 | 71.8 | 60.9 | 38.4 | 76.5 | 50.1 |
| SSR (Shen et al., 2021) | 74.2 | 54.6 | 82.5 | 60.0 | 53.2 | 29.3 | 65.6 | 35.0 |
| CSA (Ouyang et al., 2021) | 78.2 | 61.5 | 88.2 | 71.6 | 59.1 | 38.2 | 75.3 | 51.0 |
| STML (Kim et al., 2022) | 74.1 | 53.5 | 85.4 | 68.0 | 57.1 | 27.5 | 70.0 | 42.9 |
| ConAff (Yu et al., 2023) | 74.5 | 53.9 | 88.0 | 61.4 | 56.4 | 30.3 | 73.9 | 33.6 |
| **LPMT** | **84.7** | **64.8** | **93.0** | **76.1** | **67.8** | **41.4** | **84.1** | **60.1** |

can be aggregated to strengthen the local consistency. Additionally, if the neighbor satisfies the reciprocal condition, we can emphasize its importance using a parameter $\kappa$. In this case, the final neighbor-aware probability distribution $\boldsymbol{p}_i$ for each $x_i$ is given by:

$$\boldsymbol{p}_i = \sum_{j \in \mathcal{N}(i,k_2)} (\kappa \mathbb{1}^{\mathcal{R}}_{ij} + 1)\hat{\boldsymbol{p}}_j / (\kappa |\mathcal{R}(i, k_2)| + k_2), \quad (16)$$

where the element $\mathbb{1}^{\mathcal{R}}_{ij} = 1$ if $j \in \mathcal{R}(i, k_2)$, and otherwise $\mathbb{1}^{\mathcal{R}}_{ij} = 0$. The resulting probability state distributions for each instance in $\mathcal{X}$ can be organized as $\{\boldsymbol{p}_i\}_{i=1}^n \in \mathbb{R}^n$.

### 4.3. Thermodynamic Markovian Transition

Despite the ability of the obtained probability distributions to preserve local effectiveness, traditional distance metrics such as total variation fail to effectively discriminate instances outside the local region. To address this issue, we formulate a stochastic thermodynamic process to perform a Markovian transition flow over the graph, where the time evolution cost can serve as a distance for instance retrieval. For the two distributions $\boldsymbol{p}_i, \boldsymbol{p}_j$ characterizing instances $x_i, x_j$, we present a time-dependent probability distribution $\boldsymbol{q}_t$ on the graph from $\boldsymbol{q}_0 = \boldsymbol{p}_i$ to $\boldsymbol{q}_\tau = \boldsymbol{p}_j$ in a time interval $[0, \tau]$ to represent the transition flow. With $t$ representing the

continuous time variable, this dynamics evolving over the graph $\mathcal{G} = \{\mathcal{V}, \mathcal{E}\}$ represents a Markov process governed by the Langevin equations, with the corresponding discretized master equation (Seifert, 2012) expressed as:

$$\dot{\boldsymbol{q}}_t = \boldsymbol{T}_t \boldsymbol{q}_t, \quad (17)$$

where $\dot{\boldsymbol{q}}_t$ represents the derivative with respect to time. Additionally, $\boldsymbol{T}_t$ denotes the transition rate matrix, with its diagonal components satisfying $\boldsymbol{T}_t[r, r] = -\sum_{s \neq r} \boldsymbol{T}_t[s, r]$.

Since $\boldsymbol{p}_i$ and $\boldsymbol{p}_j$ may not reside within the same local region, we additionally require the long-term transition $\boldsymbol{q}_t$ to follow a path linking $\boldsymbol{p}_i$ and $\boldsymbol{p}_j$. This path comprises multiple temperate states, ensuring that each state is reachable from its predecessor only within the local region. Formally, given the set of distributions $\{\boldsymbol{p}_i\}_{i=1}^n$, the strategy $\pi$ systematically selects a sequence of temperate states $\{\boldsymbol{p}_{i_k}\}_{k=0}^K$ that satisfy local constraints. The resulting path can be expressed as $\{\boldsymbol{p}_{i_0}, \boldsymbol{p}_{i_1}, \ldots, \boldsymbol{p}_{i_K}\} = \pi(\{\boldsymbol{p}_i\}_{i=1}^n)$. Each stage of $\boldsymbol{q}_t$ spans a time interval of $\tau/K$ with $\boldsymbol{q}_{\frac{k\tau}{K}} = \boldsymbol{p}_{i_k}$. Consequently, for two given distributions $\boldsymbol{p}_i$ and $\boldsymbol{p}_j$, the minimum cost of Markovian transition flow is defined by optimizing overall potential paths $\pi$ and transition rate matrices $\boldsymbol{T}_t$ as follows:

$$d'(i, j) = \min_{\pi, \boldsymbol{T}_t} \sum_{k=0}^{K-1} \int_{\frac{k\tau}{K}}^{\frac{(k+1)\tau}{K}} \sum_{\substack{r,s=1 \\ r<s}}^n |J(r, s, t)|\, d(r, s)\mathrm{dt}, \quad (18)$$

Table 2. Evaluation of the retrieval performances based on global image features extracted by DOLG (Yang et al., 2021).

| Method | Easy | | Medium | | Hard | |
|---|---|---|---|---|---|---|
| | *R*Oxf | *R*Par | *R*Oxf | *R*Par | *R*Oxf | *R*Par |
| DOLG | 93.4 | 95.2 | 81.2 | 90.1 | 62.6 | 79.2 |
| AQE | 96.0 | 95.6 | 83.5 | 90.5 | 67.5 | 80.0 |
| $\alpha$QE | 96.7 | 95.7 | 83.9 | 91.4 | 67.6 | 81.7 |
| SG | 97.7 | 95.7 | 85.1 | 91.7 | 70.3 | 82.9 |
| STML | 97.6 | 95.4 | 86.0 | 91.5 | 70.8 | 82.3 |
| AQEwD | 97.5 | 95.6 | 84.7 | 91.2 | 68.7 | 81.1 |
| DFS | 87.3 | 93.6 | 76.1 | 90.8 | 53.5 | 82.4 |
| RDP | 95.7 | 95.0 | 87.2 | 93.0 | 72.0 | 84.8 |
| CAS | 96.8 | 95.7 | 89.5 | 93.6 | 76.7 | 86.7 |
| GSS | 98.0 | 95.3 | 86.9 | 90.6 | 72.9 | 81.2 |
| ConAff | 95.1 | 93.0 | 84.6 | 91.3 | 66.7 | 79.9 |
| LPMT | 99.7 | 95.9 | 91.4 | 95.3 | 78.2 | 89.8 |

Table 3. Evaluation of the retrieval performances based on global image features extracted by CVNet (Lee et al., 2022).

| Method | Easy | | Medium | | Hard | |
|---|---|---|---|---|---|---|
| | *R*Oxf | *R*Par | *R*Oxf | *R*Par | *R*Oxf | *R*Par |
| CVNet | 94.3 | 93.9 | 81.0 | 88.8 | 62.1 | 76.5 |
| AQE | 94.7 | 94.4 | 82.1 | 90.2 | 64.4 | 78.8 |
| $\alpha$QE | 95.8 | 94.8 | 95.8 | 90.9 | 63.5 | 80.4 |
| SG | 99.0 | 95.0 | 86.1 | 90.6 | 69.3 | 80.5 |
| STML | 98.5 | 94.9 | 86.2 | 90.8 | 69.3 | 80.5 |
| AQEwD | 96.2 | 94.9 | 84.0 | 90.8 | 66.4 | 80.0 |
| DFS | 83.5 | 93.5 | 70.8 | 89.8 | 47.4 | 79.6 |
| RDP | 96.9 | 94.5 | 87.8 | 92.4 | 71.5 | 83.3 |
| CAS | 97.6 | 95.0 | 87.6 | 92.8 | 72.7 | 84.8 |
| GSS | 99.0 | 94.0 | 87.6 | 87.1 | 70.4 | 76.9 |
| ConAff | 98.3 | 92.4 | 87.5 | 90.2 | 70.3 | 77.7 |
| LPMT | 99.5 | 95.9 | 90.2 | 94.5 | 75.2 | 88.0 |

Table 4. Evaluation of unsupervised content-based image retrieval.

| Method | CUB200 | | Indoor | | Caltech101 | |
|---|---|---|---|---|---|---|
| | mAP | R@1 | mAP | R@1 | mAP | R@1 |
| Baseline | 27.9 | 55.8 | 51.8 | 79.0 | 77.9 | 92.3 |
| AQE | 35.9 | 54.3 | 62.5 | 78.1 | 85.5 | 91.8 |
| $\alpha$QE | 35.9 | 54.8 | 62.4 | 79.1 | 85.7 | 92.5 |
| STML | 34.1 | 58.4 | 58.6 | 80.5 | 83.2 | 93.4 |
| AQEwD | 36.4 | 55.2 | 62.4 | 79.6 | 86.8 | 92.8 |
| DFS | 34.1 | 56.0 | 59.3 | 79.2 | 83.4 | 92.6 |
| RDP | 39.6 | 59.3 | 63.9 | 80.9 | 85.9 | 93.1 |
| CAS | 41.9 | 58.7 | 60.9 | 79.0 | 86.8 | 93.3 |
| ConAff | 40.7 | 57.3 | 64.2 | 79.4 | 86.8 | 93.0 |
| LPMT | 42.1 | 59.7 | 64.9 | 81.0 | 87.3 | 93.6 |

where $J(r, s, t)$ denotes the transition current from $v_r$ to $v_s$ with the cost $d(r, s)$ in Eq. (1) at time $t$, obtained by:

$$J(r, s, t) = \boldsymbol{T}_t[r, s]\boldsymbol{q}_t[s] - \boldsymbol{T}_t[s, r]\boldsymbol{q}_t[r]. \quad (19)$$

Note that at each stage, the master equation Eq. (17) of the time derivative of the flow should be satisfied. Furthermore, to facilitate a finer analysis of the transition cost, an additional power term can be applied to the distance in Eq. (18), yielding better empirical performance. Theoretically, as shown in Appendix B, in the case of microscopically reversible dynamics, the minimum flow cost is fundamentally related to the entropy production during the transition. Moreover, the time variation problem is still hard to be directly solved; therefore, under the assumption that each transition only takes place in local regions, the Wasserstein distance $\mathcal{W}_1$ can serve as a valid equivalency, followed by:

$$d'(i, j) = \min_{\pi} \sum_{k=0}^{K-1} \mathcal{W}_1(\boldsymbol{p}_{i_k}, \boldsymbol{p}_{i_{k+1}}). \quad (20)$$

To maintain the important information in Euclidean space and enhance robustness, the final distance $d^*(i, j)$ is obtained by introducing a balance weight $\theta$ to integrate $d'(i, j)$ and $d(i, j)$, as follows:

$$d^*(i, j) = \theta d(i, j) + (1 - \theta)d'(i, j). \quad (21)$$

## 5. Experiment

### 5.1. Experiment Setup

**Datasets**. To demonstrate the effectiveness of the proposed Locality Preserving Markovian Transition (LPMT) method, we conduct experiments on the revised (Radenović et al., 2018) Oxford5k (*R*Oxf) (Philbin et al., 2007) and Paris6k (*R*Par) (Philbin et al., 2008) datasets, respectively. To further evaluate performance at scale, an extra collection of one million distractor images is incorporated, forming the large-scale *R*Oxf+1M and *R*Par+1M datasets. Additionally, following the split strategy of Hu et al. (2020), we perform unsupervised content-based image retrieval on datasets like CUB200 (Wah et al., 2011), Indoor (Quattoni & Torralba, 2009), and Caltech101 (Fei-Fei et al., 2004) to identify images belonging to the same classes.

**Evaluation Metrics**. As the primary evaluation metric, we adopt the mean Average Precision (mAP) to assess the retrieval performance. For classical instance retrieval tasks, the image database is further divided into Easy (E), Medium (M), and Hard (H) categories based on difficulty levels. Given that the positive samples are limited in unsupervised content-based retrieval tasks, Recall@1 (R@1) is also reported to quantify the accuracy of the first retrieved image.

**Implementation Details**. We employ the advanced deep retrieval models, including R-GeM (Radenović et al., 2019), MAC/R-MAC (Tolias et al., 2016), DELG (Cao et al., 2020), DOLG (Yang et al., 2021), CVNet (Lee et al., 2022), and SENet (Lee et al., 2023), to extract global image features for instance retrieval. The Euclidean distance between features is the baseline measure, while refined retrieval performance is evaluated using various re-ranking methods. For unsupervised retrieval tasks aimed at identifying similar classes, we follow Hu et al. (2020) by extracting image features using the same pretrained backbone and applying the same pooling strategy to evaluate retrieval performance.

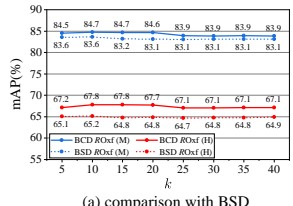
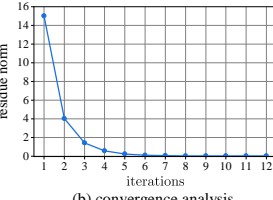

(a) comparison with BSD          (b) convergence analysis

*Figure 4.* Ablations of BCD. (a) Performance comparison with BSD. (b) Convergence analysis of BCD towards the target matrix.

## 5.2. Main Results

*Comparison of Instance Retrieval.* As summarized in Table 1, we evaluate our proposed LPMT against a wide range of re-ranking approaches, including query expansion methods (AQE, $\alpha$QE, DQE, AQEwD) *w/* and *w/o* database augmentation (DBA), diffusion-based methods (DFS, RDP, EIR, EGT, CAS), context-based methods (STML, ConAff), and learning-based methods (GSS, SSR, CSA, LAttQE), based on the global image features extracted by R-GeM. Notably, under the medium and hard evaluation protocols on $R$Oxf, LPMT improves mAP by 4.0% and 3.0% compared to the top-performing CAS, respectively. Additional results in Table 2 and Table 3 further show that LPMT consistently delivers superior performance across various retrieval models and settings. Even with high initial performance, our method can still bring further improvements, *e.g.*, from 93.4%/81.2%/62.6% to 99.7%/91.4%/78.2% on $R$Oxf with DOLG. More comparison results based on MAC/R-MAC, DELG, and SENet are shown in Appendix C; the significant improvement in performance highlights its effectiveness and robustness.

*Comparison of Unsupervised Content-based Image Retrieval.* Compared with classical instance retrieval tasks on landmark datasets, the primary challenge of this task is to find images from datasets with smaller inter-class and larger intra-class variances. Accurately identifying similar contextual neighbors is also essential for self-supervised and unsupervised learning. As shown in Table 4, LPMT achieves the mAP of 42.1%/64.9%/87.3% on CUB200, Indoor, and Caltech101, respectively, proving its effectiveness and potential for diverse machine learning applications.

## 5.3. Ablation Study

*Effectiveness of Bidirectional Collaborative Diffusion.* Bidirectional Collaborative Diffusion (BCD) is proposed to automatically integrate the diffusion processes on graphs constructed with different strategies. For a referenced nearest neighbor graph parameterized by $k$, we extend it into a graph set by scaling $k$ with factors $[1/\sqrt{2}, 1, \sqrt{2}]$ (rounded to the nearest integer) to adjust edge connectivity levels. The ablation study in Fig. 4(a) reveals that BCD delivers consistent improvements over Bidirectional Similarity Diffusion,

*Table 5.* Ablations of LSE.

| Method | $R$Oxf(M) | | | $R$Oxf(H) | | |
|---|---|---|---|---|---|---|
| | R-GeM | DOLG | CVNet | R-GeM | DOLG | CVNet |
| Baseline | 67.3 | 81.2 | 81.0 | 44.2 | 62.6 | 62.1 |
| Cosine+$k$-nn | 75.8 | 86.1 | 86.4 | 54.9 | 70.4 | 69.5 |
| Gaussian+$k$-nn | 78.2 | 88.8 | 88.7 | 58.9 | 75.5 | 73.7 |
| BSD+$k$-nn | 81.6 | 90.4 | 89.5 | 63.0 | 76.9 | 74.9 |
| BCD+$k$-nn | **84.2** | **91.3** | **90.1** | **66.1** | **78.1** | **75.2** |
| Cosine+$k$-recip | 80.6 | 87.0 | 87.5 | 61.3 | 73.7 | 70.1 |
| Gaussian+$k$-recip | 81.9 | 89.5 | 88.8 | 63.1 | 75.8 | 73.1 |
| BSD+$k$-recip | 83.6 | 90.3 | 89.5 | 65.2 | 77.2 | 73.8 |
| BCD+$k$-recip | **84.7** | **91.4** | **90.2** | **67.8** | **78.2** | **75.2** |

*Table 6.* Ablations of TMT.

| Method | $R$Oxf(M) | | | $R$Oxf(H) | | |
|---|---|---|---|---|---|---|
| | R-GeM | DOLG | CVNet | R-GeM | DOLG | CVNet |
| Baseline | 67.3 | 81.2 | 81.0 | 44.2 | 62.6 | 62.1 |
| Cosine | 78.0 | 86.9 | 85.3 | 60.0 | 72.7 | 69.3 |
| Euclidean | 78.5 | 87.3 | 85.6 | 60.7 | 73.2 | 69.5 |
| Jaccard | 78.8 | 87.4 | 85.9 | 60.5 | 74.1 | 70.6 |
| Total Variation | 79.5 | 87.8 | 86.4 | 61.7 | 74.5 | 71.1 |
| TMT | **84.7** | **91.4** | **90.2** | **67.8** | **78.2** | **75.2** |

particularly under challenging scenarios. For example, performance improves from 83.6% to 84.7% in $R$Oxf(M) and 65.2% to 67.8% in $R$Oxf(H) when $k = 10$. These improvements underscore the effectiveness of BCD in mitigating the negative impact of inappropriate connections by synthesizing information from different graphs. Furthermore, in our implementation of BCD, we perform the updating strategy of Eq. (9) for three iterations, followed by a refreshing of the weight set $\beta$ in each cycle. As shown in Fig. 4(b), the residue norm of the target similarity matrix decreases rapidly within a few iterations, demonstrating its efficiency.

*Effectiveness of Locality State Embedding.* As depicted in Section 4.2, the proposed LSE aims to encode each instance into a manifold-aware distribution using the obtained BCD similarity matrix and $k$-reciprocal strategy. To validate its effectiveness, we conduct experiments to test various combinations of Cosine (Bai & Bai, 2016), Gaussian kernel, BSD (Luo et al., 2024), and our BCD similarity, along with the embedding strategies based on $k$-nn and $k$-reciprcal. Table 5 reveals that LSE (BCD+$k$-recip) consistently outperforms other methods across diverse deep retrieval models, particularly under the hard protocol. Notably, with the same $k$-reciprocal strategy, LSE surpasses Cosine, Gaussian, and BSD by 6.5%/4.7%/2.6% on R-GeM $R$Oxf(H), highlighting its superiority in exploiting manifold information.

*Effectiveness of Thermodynamic Markovian Transition.* As discussed in Section 4.3, the minimum transition flow cost between distinct distributions is an effective distance measure for instance retrieval. To verify this, we benchmark it against common metrics such as Cosine, Euclidean, Jaccard, and Total Variation distance, with results detailed in Table 6. Under the $R$Oxf(M) and $R$Oxf(H) protocols, LPMT achieves superior performance, surpassing other metrics by

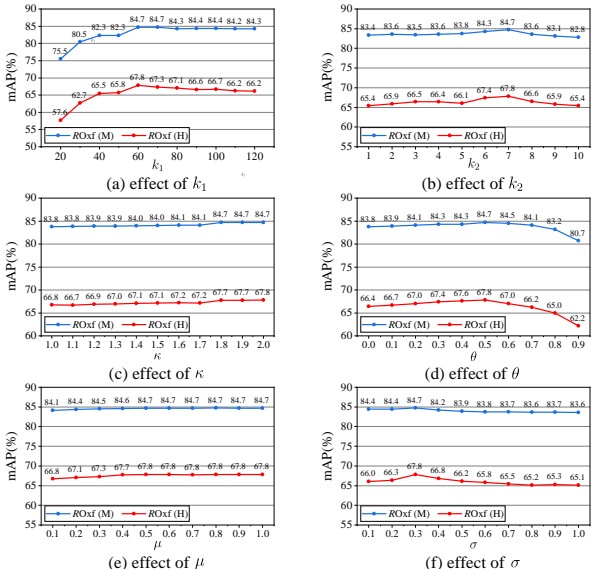

*Figure 5.* Sensitivity analysis of hyper-parameters based on image features extracted by R-GeM. (a) Effect of $k_1$. (b) Effect of $k_2$. (c) Effect of $\kappa$. (d) Effect of $\theta$. (e) Effect of $\mu$. (f) Effect of $\sigma$.

6.7%/6.2%/5.9%/5.2% and 7.8%/7.1%/7.2%/6.1% based on R-GeM. This suggests that the time evolution process can effectively capture the underlying manifold structure, benefiting the overall global retrieval performance.

*Time Complexity Analysis.* LPMT consists of three core components: BCD, LSE, and TMT, with the computational cost primarily driven by BCD and TMT. Specifically, BCD optimizes for the robust similarity matrix via a two-step iterative approach following Algorithm 1, which has a time complexity of $\mathcal{O}(n^3)$. As for TMT, we introduce an entropy regularization term in Appendix B, resulting in a fixed-point iterative solution with $\mathcal{O}(n^3)$ complexity. Consequently, the overall complexity of LPMT is $\mathcal{O}(n^3)$. To improve efficiency in practical scenarios, we re-rank only the top-$k$ images, reducing the complexity to $\mathcal{O}(k^3)$, and the execution time remains under 3 seconds when $k = 5000$.

*Effect of $k_1$ and $k_2$.* The hyper-parameters $k_1$ and $k_2$ introduced in LSE determine the size of the local region and the number of confident neighborhoods. As shown in Fig. 5(a), retrieval performance peaks at $k_1 = 60$, suggesting that selecting a moderate region size is crucial to incorporate sufficient informative instances. Similarly, Fig. 5(b) shows that $k_2 = 7$ yields optimal performance, highlighting the importance of balancing neighborhood size and the proportion of correct samples for improved representation.

*Sensitivity of Hyper-parameters.* In Eq. (16), the reciprocal neighbors enhance the LSE distribution, controlled by a hyper-parameter $\kappa$. As illustrated in Fig. 5(c), performance increases with $\kappa$ and reaches its maximum at $\kappa = 2$. Meanwhile, the hyper-parameter $\theta$ serves as a balancing weight

to fuse the original Euclidean distance with the thermodynamic transition flow cost. Fig. 5(d) reveals that $\theta = 0.5$ yields the optimal result, demonstrating that incorporating the original distance enhances the retrieval robustness. Additional analyses of hyper-parameters such as $\mu$ and $\sigma$ are provided in Fig. 5(e) and (f), confirming the robustness of our approach towards their variations.

# 6. Conclusion

To address the problem of decaying positive information and the influence of disconnections during the diffusion process, we propose a novel Locality Preserving Markovian Transition (LPMT) framework for instance retrieval. LPMT embeds each instance into a probability distribution within the data manifold and then derives a long-term stochastic thermodynamic transition process to transport the distributions along the graph, with each stage constrained within a local region. The minimum transition flow cost can maintain the local effectiveness while capturing the underlying manifold structure, serving as an effective distance measure. Extensive evaluations on several benchmarks validate the consistently superior performance of LPMT, indicating its effectiveness for instance retrieval and potential to adapt to various unsupervised machine learning models.

# Acknowledgements

This work was supported by the National Science and Technology Major Project (2021ZD0112202), the National Natural Science Foundation of China (62376268, 62476199, 62202331, U23A20387, 62036012, U21B2044).

# Impact Statement

This paper presents work whose goal is to advance the field of Machine Learning. There are many potential societal consequences of our work, none of which we feel must be specifically highlighted here.

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

# Locality Preserving Markovian Transition for Instance Retrieval

## A. Bidirectional Collaborative Diffusion

Given a reference affinity graph with its adjacency matrix denoted by $\boldsymbol{W}$, we can extend it into a graph set with the corresponding adjacency matrices $\{\boldsymbol{W}^v\}_{v=1}^m$ by adding and removing some connections. Inspired by prior work (Zhou et al., 2012; Bai et al., 2019a;b;c), we aim to integrate the Bidirectional Similarity Diffusion (Luo et al., 2024) process across these individual graphs to automatically generate a robust similarity matrix. Such that the resulting matrix can not only effectively capture the underlying manifold structure, but also mitigate the negative impact caused by suboptimal graph construction strategies. To this end, we associate each adjacency matrix $\boldsymbol{W}^v$ with a learnable aggregation weight $\beta_v$, collectively forming the weight set $\{\beta_v\}_{v=1}^m$. Additionally, we impose a normalization constraint such that the summation of $\beta_v$ equal to 1, *i.e.*, $\sum_{v=1}^m \beta_v = 1$. Under this framework, our proposed Bidirectional Collaborative Diffusion can then be formally expressed as:

$$\min_{\boldsymbol{F},\beta} \quad \sum_{v=1}^m \beta_v H^v + \frac{1}{2}\lambda\|\beta\|_2^2$$
$$\text{s.t.} \quad 0 \le \beta_v \le 1, \sum_{v=1}^m \beta_v = 1, \tag{A.1}$$

where $H^v$ denotes the Bidirectional Similarity Diffusion process (Luo et al., 2024) with its objective function follows as:

$$H^v = \frac{1}{4}\sum_{k=1}^n \sum_{i,j=1}^n \boldsymbol{W}_{ij}^v \Big(\frac{\boldsymbol{F}_{ki}}{\sqrt{\boldsymbol{D}_{ii}^v}} - \frac{\boldsymbol{F}_{kj}}{\sqrt{\boldsymbol{D}_{jj}^v}}\Big)^2 + \boldsymbol{W}_{ij}^v \Big(\frac{\boldsymbol{F}_{ik}}{\sqrt{\boldsymbol{D}_{ii}^v}} - \frac{\boldsymbol{F}_{jk}}{\sqrt{\boldsymbol{D}_{jj}^v}}\Big)^2 + \mu\|\boldsymbol{F} - \boldsymbol{E}\|_F^2. \tag{A.2}$$

For each adjacency matrix $\boldsymbol{W}^v$, $\boldsymbol{D}^v$ is the corresponding diagonal matrix with its $i$-th diagonal element equal to the summation of the $i$-th row in $\boldsymbol{W}^v$. The regularization term is weighted by a positive constant $\mu > 0$, and $\boldsymbol{E}$ is a positive semi-definite matrix introduced to prevent $\boldsymbol{F}$ from being extremely smooth. Given that the regularization term is shared across all $H^v$, we can factor it out to derive an equivalent optimization problem, thereby simplifying the subsequent analysis:

$$\min_{\boldsymbol{F},\beta} \quad \sum_{v=1}^m \beta_v \tilde{H}^v + \mu\|\boldsymbol{F} - \boldsymbol{E}\|_F^2 + \frac{1}{2}\lambda\|\beta\|_2^2$$
$$\text{s.t.} \quad 0 \le \beta_v \le 1, \sum_{v=1}^m \beta_v = 1, \tag{A.3}$$

where

$$\tilde{H}^v = \frac{1}{4}\sum_{k=1}^n \sum_{i,j=1}^n \boldsymbol{W}_{ij}^v \Big(\frac{\boldsymbol{F}_{ki}}{\sqrt{\boldsymbol{D}_{ii}^v}} - \frac{\boldsymbol{F}_{kj}}{\sqrt{\boldsymbol{D}_{jj}^v}}\Big)^2 + \boldsymbol{W}_{ij}^v \Big(\frac{\boldsymbol{F}_{ik}}{\sqrt{\boldsymbol{D}_{ii}^v}} - \frac{\boldsymbol{F}_{jk}}{\sqrt{\boldsymbol{D}_{jj}^v}}\Big)^2. \tag{A.4}$$

The above optimization problem involves both $\beta$ and $\boldsymbol{F}$, making it inherently complex and impractical to solve directly. To resolve this, we propose a numerical approach that decomposes the objective function into two subproblems, allowing for a systematic iterative approximation of the optimal result by recursively optimizing $\boldsymbol{F}$ and $\beta$. In the following, we first describe the update of $\boldsymbol{F}$ with fixed $\beta$ in Section A.1, followed by the update of $\beta$ with fixed $\boldsymbol{F}$ in Section A.2. Finally, in Section A.3, we introduce a fixed-point iteration scheme to progressively approach the optimal solution.

### A.1. Optimize $\boldsymbol{F}$ with Fixed $\beta$

In the case when $\{\beta_v\}_{v=1}^m$ are fixed, such that the constraint of $\|\beta\|_2^2$ can be omitted when finding the optimal value of $\boldsymbol{F}$, bring Eq. (A.4) into the target function, the optimization problem can be rewritten into:

$$\min_{\boldsymbol{F}} \frac{1}{4}\sum_{v=1}^m \sum_{k=1}^n \sum_{i,j=1}^n \beta_v \Big(\boldsymbol{W}_{ij}^v \Big(\frac{\boldsymbol{F}_{ki}}{\sqrt{\boldsymbol{D}_{ii}^v}} - \frac{\boldsymbol{F}_{kj}}{\sqrt{\boldsymbol{D}_{jj}^v}}\Big)^2 + \boldsymbol{W}_{ij}^v \Big(\frac{\boldsymbol{F}_{ik}}{\sqrt{\boldsymbol{D}_{ii}^v}} - \frac{\boldsymbol{F}_{jk}}{\sqrt{\boldsymbol{D}_{jj}^v}}\Big)^2\Big) + \mu\|\boldsymbol{F} - \boldsymbol{E}\|_F^2. \tag{A.5}$$

The matrix-based formulation is still difficult to deal with. In the following, we will first reformulate it into a vectorized form to simplify the derivation of the closed-form solution, and then demonstrate that the optimal solution can be iteratively approximated with reduced computational cost.

*Vectorized Formulation.* To facilitate the vectorized transformation, we first introduce an identity matrix $I$ into Eq. (A.4), such that $\tilde{H}^v$ can be rewritten into:

$$\tilde{H}^v = \frac{1}{4} \sum_{k,l=1}^{n} \sum_{i,j=1}^{n} \left( \boldsymbol{W}_{ij} \boldsymbol{I}_{kl} \left( \frac{\boldsymbol{F}_{ki}}{\sqrt{\boldsymbol{D}_{ii}}} - \frac{\boldsymbol{F}_{lj}}{\sqrt{\boldsymbol{D}_{jj}}} \right)^2 + \boldsymbol{I}_{kl} \boldsymbol{W}_{ij} \left( \frac{\boldsymbol{F}_{ik}}{\sqrt{\boldsymbol{D}_{ii}}} - \frac{\boldsymbol{F}_{jl}}{\sqrt{\boldsymbol{D}_{jj}}} \right)^2 \right). \tag{A.6}$$

Afterwards, we introduce the vectorization operator $vec(\cdot)$, which can stack the columns in a matrix one after another to formulate a single column vector, and the Kronecker product $\otimes$, which combines two matrices to produce a new one. By taking advantage of these two transformations, we proceed to define the Kronecker product $\mathbb{W}^{v(1)} = \boldsymbol{W}^v \otimes \boldsymbol{I}$, $\mathbb{D}^{v(1)} = \boldsymbol{D}^v \otimes \boldsymbol{I}$ for the former part, and $\mathbb{W}^{v(2)} = \boldsymbol{I} \otimes \boldsymbol{W}^v$, $\mathbb{D}^{v(2)} = \boldsymbol{I} \otimes \boldsymbol{D}^v$ for the latter part. The corresponding items between the original and Kronecker formation are associated with the newly defined corner markers $p \equiv n(i - 1) + k$, $q \equiv n(j - 1) + l$, $r \equiv n(k - 1) + i$, and $s \equiv n(l - 1) + j$. In addition, define the normalized matrix of $\boldsymbol{W}^v$ as $\boldsymbol{S}^v = (\boldsymbol{D}^v)^{-1/2} \boldsymbol{W}^v (\boldsymbol{D}^v)^{-1/2}$, $\mathbb{S}^{v(1)} = \boldsymbol{S}^v \otimes \boldsymbol{I}$ and $\mathbb{S}^{v(2)} = \boldsymbol{I} \otimes \boldsymbol{S}^v$. The following facts can be easily established:

1. $vec(\boldsymbol{F})_p = \boldsymbol{F}_{ki}$ and $vec(\boldsymbol{F})_q = \boldsymbol{F}_{lj}$; $vec(\boldsymbol{F})_r = \boldsymbol{F}_{ik}$ and $vec(\boldsymbol{F})_s = \boldsymbol{F}_{jl}$.

2. $\mathbb{W}_{pq}^{v(1)} = \boldsymbol{W}_{ij}^v \boldsymbol{I}_{kl}$, $\mathbb{D}_{pp}^{v(1)} = \boldsymbol{D}_{ii}^v$ and $\mathbb{D}_{qq}^{v(1)} = \boldsymbol{D}_{jj}^v$; $\mathbb{W}_{rs}^{v(2)} = \boldsymbol{I}_{kl} \boldsymbol{W}_{ij}$, $\mathbb{D}_{rr}^{v(2)} = \boldsymbol{D}_{ii}^v$ and $\mathbb{D}_{ss}^{v(2)} = \boldsymbol{D}_{jj}^v$.

3. $\sum_{q=1}^{n^2} \mathbb{W}_{pq}^{v(1)} = \mathbb{D}_{pp}^{v(1)}$ and $\sum_{p=1}^{n^2} \mathbb{W}_{pq}^{v(1)} = \mathbb{D}_{qq}^{v(1)}$; $\sum_{s=1}^{n^2} \mathbb{W}_{rs}^{v(2)} = \mathbb{D}_{rr}^{v(2)}$ and $\sum_{r=1}^{n^2} \mathbb{W}_{rs}^{v(2)} = \mathbb{D}_{ss}^{v(2)}$.

4. $\mathbb{S}^{v(1)} = (\mathbb{D}^{v(1)})^{-1/2} \mathbb{W}^{v(1)} (\mathbb{D}^{v(1)})^{-1/2}$ and $\mathbb{S}^{v(2)} = (\mathbb{D}^{v(2)})^{-1/2} \mathbb{W}^{v(2)} (\mathbb{D}^{v(2)})^{-1/2}$.

By substituting the above transformations into Eq. (A.6), the representation of $\tilde{H}^v$ can be equivalently reformulated in the Kronecker product form as follows:

$$\begin{aligned} \tilde{H}^v =& \frac{1}{4} \sum_{p,q=1}^{n^2} \mathbb{W}_{pq}^{v(1)} \left( \frac{vec(\boldsymbol{F})_p}{\sqrt{\mathbb{D}_{pp}^{v(1)}}} - \frac{vec(\boldsymbol{F})_q}{\sqrt{\mathbb{D}_{qq}^{v(1)}}} \right)^2 + \frac{1}{4} \sum_{r,s=1}^{n^2} \mathbb{W}_{rs}^{v(2)} \left( \frac{vec(\boldsymbol{F})_r}{\sqrt{\mathbb{D}_{rr}^{v(2)}}} - \frac{vec(\boldsymbol{F})_s}{\sqrt{\mathbb{D}_{ss}^{v(2)}}} \right)^2 \\ =& vec(\boldsymbol{F})^\top \left( \mathbb{I} - \frac{1}{2} (\mathbb{D}^{v(1)})^{-1/2} \mathbb{W}^{v(1)} (\mathbb{D}^{v(1)})^{-1/2} - \frac{1}{2} (\mathbb{D}^{v(2)})^{-1/2} \mathbb{W}^{v(2)} (\mathbb{D}^{v(2)})^{-1/2} \right) vec(\boldsymbol{F}) \\ =& vec(\boldsymbol{F})^\top \left( \mathbb{I} - \frac{1}{2} \mathbb{S}^{v(1)} - \frac{1}{2} \mathbb{S}^{v(2)} \right) vec(\boldsymbol{F}). \end{aligned} \tag{A.7}$$

Basically, the Frobenius norm of $\boldsymbol{F} - \boldsymbol{E}$ within the regularization term is equivalent to the $L_2$-norm of $vec(\boldsymbol{F} - \boldsymbol{E})$, combine the Kronecker matrix based $\tilde{H}^v$ and the regularization term together, the objective function Eq. (A.5) can be rewritten into:

$$\min_{\boldsymbol{F}} \sum_{v=1}^{m} \beta_v H^v = \min_{\boldsymbol{F}} \sum_{v=1}^{m} \beta_v vec(\boldsymbol{F})^\top \left( \mathbb{I} - \frac{1}{2} \mathbb{S}^{v(1)} - \frac{1}{2} \mathbb{S}^{v(2)} \right) vec(\boldsymbol{F}) + \mu \| vec(\boldsymbol{F} - \boldsymbol{E}) \|_2^2. \tag{A.8}$$

**Lemma A.1.** *Let $\boldsymbol{A} \in \mathbb{R}^{n \times n}$, the spectral radius of $\boldsymbol{A}$ is denoted as $\rho(\boldsymbol{A}) = \max\{|\lambda|, \lambda \in \sigma(\boldsymbol{A})\}$, where $\sigma(\boldsymbol{A})$ is the spectrum of $\boldsymbol{A}$ that represents the set of all the eigenvalues. Let $\| \cdot \|$ be a matrix norm on $\mathbb{R}^{n \times n}$, given a square matrix $\boldsymbol{A} \in \mathbb{R}^{n \times n}$, $\lambda$ is an arbitrary eigenvalue of $\boldsymbol{A}$, then we have $|\lambda| \leq \rho(\boldsymbol{A}) \leq \|\boldsymbol{A}\|$.*

**Lemma A.2.** *Let $\boldsymbol{A} \in \mathbb{R}^{m \times m}$, $\boldsymbol{B} \in \mathbb{R}^{n \times n}$, denote $\{\lambda_i, \boldsymbol{x}_i\}_{i=1}^m$ and $\{\mu_i, \boldsymbol{y}_i\}_{i=1}^n$ as the eigen-pairs of $\boldsymbol{A}$ and $\boldsymbol{B}$ respectively. The set of $mn$ eigen-pairs of $\boldsymbol{A} \otimes \boldsymbol{B}$ is given by:*

$$\{\lambda_i \mu_j, \boldsymbol{x}_i \otimes \boldsymbol{y}_j\}_{i=1,\dots,m,\ j=1,\dots n}.$$

*Closed-form Solution.* Suppose the objective function in Eq. (A.8) that needs to be minimized is $J$. To prove that $J$ is convex, it is equivalent to show that its Hessian matrix is positive. To get started, we first consider the matrix $(\boldsymbol{D}^v)^{-1} \boldsymbol{W}^v$, whose induced $l_\infty$-norm is equal to 1, *i.e.*, $\|(\boldsymbol{D}^v)^{-1} \boldsymbol{W}^v\|_\infty = 1$, since the $i$-th diagonal element in matrix $\boldsymbol{D}^v$ equal to the summation of the corresponding $i$-th row in matrix $\boldsymbol{W}^v$. Lemma A.1 gives that $\rho((\boldsymbol{D}^v)^{-1} \boldsymbol{W}^v) \leq 1$. As for the

matrix $\boldsymbol{S}^v = (\boldsymbol{D}^v)^{-1/2}\boldsymbol{W}^v(\boldsymbol{D}^v)^{-1/2}$ we are concerned about, since we can rewrite it as $(\boldsymbol{D}^v)^{1/2}(\boldsymbol{D}^v)^{-1}\boldsymbol{W}^v(\boldsymbol{D}^v)^{-1/2}$, thus it is similar to $(\boldsymbol{D}^v)^{-1}\boldsymbol{W}^v$, *i.e.*, $\boldsymbol{S}^v \sim (\boldsymbol{D}^v)^{-1}\boldsymbol{W}^v$. This implies that the two matrices share the same eigenvalues, such that $\rho(\boldsymbol{S}^v) \leq 1$. By applying Lemma A.2, we can conclude that both the spectral radius of the Kronecker product $\mathbb{S}^{v(1)} = \boldsymbol{S}^v \otimes \boldsymbol{I}$ and $\mathbb{S}^{v(2)} = \boldsymbol{I} \otimes \boldsymbol{S}^v$ is no larger than 1, *i.e.*, $\rho(\mathbb{S}^{v(1)}) \leq 1, \rho(\mathbb{S}^{v(2)}) \leq 1$.

The Hessian matrix of Eq. (A.8) can be obtained as $2(\mu+1)\mathbb{I} - \sum_{v=1}^m \beta_v(\bar{\mathbb{S}}^{v(1)} + \bar{\mathbb{S}}^{v(2)})$, where $2\bar{\mathbb{S}}^{v(1)} = \mathbb{S}^{v(1)} + (\mathbb{S}^{v(1)})^\top$ and $2\bar{\mathbb{S}}^{v(2)} = \mathbb{S}^{v(2)} + (\mathbb{S}^{v(2)})^\top$. Given that $\mu > 0$, $\sum_{v=1}^m \beta_v = 1$, and each $\rho(\mathbb{S}^v) \leq 1$, such that the eigenvalues are greater than 0, indicating that the Hessian matrix is positive-definite and the objective function $J$ is convex. Consequently, we can take the partial derivative of $vec(\boldsymbol{F})$ to obtain the optimal result of Eq. (A.8), followed by:

$$\nabla_{vec(\boldsymbol{F})} J = \sum_{v=1}^M \beta_v(2\mathbb{I} - \bar{\mathbb{S}}^{v(1)} - \bar{\mathbb{S}}^{v(2)})vec(\boldsymbol{F}) + 2\mu(vec(\boldsymbol{F} - \boldsymbol{E})). \tag{A.9}$$

The optimal solution $\boldsymbol{F}^*$ is found by setting the above partial derivative to zero, yielding:

$$vec(\boldsymbol{F}^*) = \frac{2\mu}{\mu+1}\Big(2\mathbb{I} - \sum_{v=1}^m \frac{\beta_v}{\mu+1}\bar{\mathbb{S}}^{v(1)} - \sum_{v=1}^m \frac{\beta_v}{\mu+1}\bar{\mathbb{S}}^{v(2)}\Big)^{-1} vec(\boldsymbol{E}). \tag{A.10}$$

To get a simpler representation, we substitute $\alpha_v$ with $\frac{\beta_v}{\mu+1}$, $\bar{\mathbb{S}}^v$ with $(\bar{\mathbb{S}}^{v(1)} + \bar{\mathbb{S}}^{v(2)})/2$, and denote $\alpha = \sum_{v=1}^m \alpha_v$, resulting in the closed-form solution of the optimal $\boldsymbol{F}^*$ when $\{\beta_v\}_{v=1}^m$ are fixed, which can be expressed as:

$$\boldsymbol{F}^* = (1-\alpha)vec^{-1}\big((\mathbb{I} - \sum_{v=1}^m \alpha_v\bar{\mathbb{S}}^v)^{-1}vec(\boldsymbol{E})\big). \tag{A.11}$$

**Lemma A.3.** *Let $\boldsymbol{A} \in \mathbb{R}^{m \times n}$, $\boldsymbol{X} \in \mathbb{R}^{n \times p}$ and $\boldsymbol{B} \in \mathbb{R}^{p \times q}$ respectively, then*

$$vec(\boldsymbol{AXB}) = (\boldsymbol{B}^\top \otimes \boldsymbol{A})vec(\boldsymbol{X}).$$

**Lemma A.4.** *Let $\boldsymbol{A} \in \mathbb{R}^{n \times n}$, then $\lim_{k \to \infty} \boldsymbol{A}^k = 0$ if and only if $\rho(\boldsymbol{A}) < 1$.*

**Lemma A.5.** *Given a matrix $\boldsymbol{A} \in \mathbb{R}^{n \times n}$ and $\rho(\boldsymbol{A}) < 1$, the Neumann series $\boldsymbol{I} + \boldsymbol{A} + \boldsymbol{A}^2 + \cdots$ converges to $(\boldsymbol{I} - \boldsymbol{A})^{-1}$.*

*Iterative Solution.* Utilizing the relationship given by Lemma A.3, we could put all the matrices in Eq. (A.9) into the $vec(\cdot)$ operator. Additionally, setting the derivative to 0, we can obtain the following equivalent relationship:

$$2\boldsymbol{F} - \sum_{v=1}^m \beta_v\boldsymbol{F}\bar{\boldsymbol{S}}^v - \sum_{v=1}^m \beta_v\bar{\boldsymbol{S}}^v\boldsymbol{F} + 2\mu(\boldsymbol{F} - \boldsymbol{E}) = 0. \tag{A.12}$$

By making some small changes to the above formula, the optimum result $\boldsymbol{F}^*$ is actually the solution to the following Lyapunov equation:

$$(\boldsymbol{I} - \sum_{v=1}^m \alpha_v\bar{\boldsymbol{S}}^v)\boldsymbol{F} + \boldsymbol{F}(\boldsymbol{I} - \sum_{v=1}^m \alpha_v\bar{\boldsymbol{S}}^v) = 2(1-\alpha)\boldsymbol{E}. \tag{A.13}$$

Directly solving this equation incurs a significantly high time complexity, but we can approximate the optimal solution at a lower cost in an iterative manner. Inspired by (Zhou et al., 2003; Iscen et al., 2017; Bai et al., 2017a; Luo et al., 2024), we can develop an iterative function to infinitely approach the optimal result as follows:

$$\boldsymbol{F}^{(t+1)} = \frac{1}{2}\sum_{v=1}^m \alpha_v\big(\boldsymbol{F}^{(t)}(\bar{\boldsymbol{S}}^v)^\top + \bar{\boldsymbol{S}}^v\boldsymbol{F}^{(t)}\big) + (1-\alpha)\boldsymbol{E}, \tag{A.14}$$

where $\boldsymbol{S}^v = (\boldsymbol{D}^v)^{-1/2}\boldsymbol{W}^v(\boldsymbol{D}^v)^{-1/2}$, and $\bar{\boldsymbol{S}}^v = (\boldsymbol{S}^v + (\boldsymbol{S}^v)^\top)/2$. By applying Lemma A.3, we can add a $vec(\cdot)$ operator on both left-hand and right-hand sides, reformulating the iteration process as:

$$\begin{aligned}
vec(\boldsymbol{F}^{(t+1)}) &= \frac{1}{2}\sum_{v=1}^m \alpha_v(\bar{\boldsymbol{S}}^v \otimes \boldsymbol{I} + \boldsymbol{I} \otimes \bar{\boldsymbol{S}}^v)vec(F^{(t)}) + (1-\alpha)vec(\boldsymbol{E}) \\
&= \sum_{v=1}^m \alpha_v\bar{\mathbb{S}}^v vec(\boldsymbol{F}^{(t)}) + (1-\alpha)vec(\boldsymbol{E}).
\end{aligned} \tag{A.15}$$

---

**Algorithm 2** Effective Solution of Eq. (A.13)

---

**Input:** Adjacency matrix set $\{\boldsymbol{W}^v\}_{v=1}^m$, initial estimation of similarity matrix $\boldsymbol{F}^{(0)}$, normalized Kronecker matrix $\{\bar{\mathbb{S}}^v\}_{v=1}^m$, identity matrix $\mathbb{I} \in \mathbb{R}^{n^2 \times n^2}$, max number of iterations $maxiter$, hyper-parameter $\mu$, $\lambda$, iteration tolerance $\delta$.

**Output:** $\boldsymbol{F}^* = vec^{-1}(\boldsymbol{f}^*)$.

 1: substitute $\sum_{v=1}^m \alpha_v \bar{\boldsymbol{S}}^v$ with $\bar{\boldsymbol{S}}$, and $\sum_{v=1}^m \alpha_v \bar{\mathbb{S}}^v$ with $\bar{\mathbb{S}}$
 2: initialize $\boldsymbol{P}^{(0)}$ and $\boldsymbol{R}^{(0)}$ with $2(1-\alpha)\boldsymbol{E} - (\boldsymbol{I} - \bar{\boldsymbol{S}})\boldsymbol{F}^{(0)} - \boldsymbol{F}^{(0)}(\boldsymbol{I} - \alpha\bar{\boldsymbol{S}})$
 3: denote $\boldsymbol{f}_t = vec(\boldsymbol{F}^{(t)})$, $\boldsymbol{r}_t = vec(\boldsymbol{R}^{(t)})$, $\boldsymbol{p}_t = vec(\boldsymbol{P}^{(t)})$
 4: **repeat**
 5:     compute $\alpha_t = \dfrac{\boldsymbol{r}_t^\top \boldsymbol{r}_t}{2\boldsymbol{p}_t^\top (\mathbb{I} - \alpha\bar{\mathbb{S}})\boldsymbol{p}_t}$
 6:     refresh $\boldsymbol{f}_{t+1} = \boldsymbol{f}_t + \alpha_t \boldsymbol{p}_t$
 7:     update $\boldsymbol{r}_{t+1} = \boldsymbol{r}_t - 2\alpha_t(\mathbb{I} - \alpha\bar{\mathbb{S}})\boldsymbol{p}_t$
 8:     compute $\beta_t = \dfrac{\boldsymbol{r}_{t+1}^\top \boldsymbol{r}_{t+1}}{\boldsymbol{r}_t^\top \boldsymbol{r}_t}$
 9:     refresh $\boldsymbol{p}_{t+1} = \boldsymbol{r}_{t+1} + \beta_t \boldsymbol{p}_t$
10:     $t \leftarrow t + 1$
11: **until** $t = maxiter$ or $\|\boldsymbol{r}_{t+1}\| < \delta$

---

Assume the iteration starts with an initial value $\boldsymbol{F}^{(0)}$, which can be chosen as either the diagonal matrix $\boldsymbol{I}$ or the regularization matrix $\boldsymbol{E}$. Through iteratively applying the update rule, we derive an expression in which $\boldsymbol{F}^{(t+1)}$ is explicitly formulated in terms of $\boldsymbol{F}^{(0)}$, the normalized matrix $\bar{\mathbb{S}}$, and the regularization matrix $\boldsymbol{E}$ without any direct dependence on the immediate previous value $\boldsymbol{F}^{(t)}$, following:

$$vec(\boldsymbol{F}^{(t)}) = \sum_{v=1}^m (\alpha_v \bar{\mathbb{S}}^v)^t vec(\boldsymbol{F}^{(0)}) + (1 - \sum_{v=1}^m \alpha_v) \sum_{i=0}^{t-1} (\sum_{v=1}^m \alpha_v \bar{\mathbb{S}}^v)^i vec(\boldsymbol{E}). \tag{A.16}$$

Since we have already proved that the spectral radius of $\bar{\mathbb{S}}^v$ is no larger than 1, by taking advantage of Lemma A.4 and A.5, we can easily demonstrate that the following two expressions hold true:

$$\lim_{t \to \infty} \sum_{v=1}^m (\alpha_v \bar{\mathbb{S}}^v)^t = 0, \tag{A.17}$$

$$\lim_{t \to \infty} \sum_{i=0}^{t-1} (\sum_{v=1}^m \alpha_v \bar{\mathbb{S}}^v)^i = (\mathbb{I} - \sum_{v=1}^m \alpha_v \bar{\mathbb{S}}^v)^{-1}. \tag{A.18}$$

Therefore, the iterative sequence of $vec(\boldsymbol{F}^{(t)})$ asymptotically approaches a stable solution, converging to:

$$vec(F^*) = (1 - \alpha)(\mathbb{I} - \sum_{v=1}^m \alpha_v \bar{\mathbb{S}}^v)^{-1} vec(E). \tag{A.19}$$

By performing the inverse operator $vec^{-1}(\cdot)$ on both side, the optimal result for Eq. (A.5) can be derived, which yields:

$$\boldsymbol{F}^* = (1 - \alpha) vec^{-1}\big((\mathbb{I} - \sum_{v=1}^m \alpha_v \bar{\mathbb{S}}^v)^{-1} vec(\boldsymbol{E})\big). \tag{A.20}$$

The above expression is identical to Eq. (A.11), which implies that the time complexity of solving the Lyapunov equation in Eq. (A.13) can be receded to $\mathcal{O}(n^3)$, where $n$ denotes the dimension of the matrix. Inspired by Iscen et al. (2017; 2018); Luo et al. (2024), the convergence rate can be further accelerated with the conjugate gradient method. In other words, the solution to the equation can be estimated with fewer iterations following Algorithm 2. Specifically, starting from an initial estimation $\boldsymbol{F}^{(0)}$, the iteration in the Bidirectional Collaborative Diffusion process will cease when the maximum count $maxiter$ is reached or the norm of the residue is less than a predefined tolerance $\delta$.

**A.2. Optimize $\beta$ with fixed $F$**

When $F$ is fixed, the objective value of $H^v$ for each adjacency matrix $\boldsymbol{W}^v$ in Eq. (A.2) can be directly computed. As a result, the optimization of $\beta$ reduces to solving the following problem:

$$\min_{\beta} \quad \sum_{v=1}^{m} \beta_v H^v + \frac{1}{2}\lambda\|\beta\|_2^2$$
$$\text{s.t.} \quad 0 \leq \beta_v \leq 1, \sum_{v=1}^{m} \beta_v = 1. \tag{A.21}$$

Specifically, the objective function in Eq. (A.21) takes the form of a Lasso optimization problem, which can be solved by utilizing the coordinate descent method following (Bai et al., 2017b; 2019c), as demonstrated below:

$$\begin{cases} \beta_i^* = \frac{\lambda(\beta_i+\beta_j)+(H^j-H^i)}{2\lambda}, \\ \beta_j^* = \beta_i + \beta_j - \beta_i^*. \end{cases} \tag{A.22}$$

During the updating procedure, both $\beta_i$ and $\beta_j$ should not violate the inequality constraint $0 \leq \beta_v \leq 1$. To achieve this, we explicitly set $\beta_i^*$ to be zero if $\lambda(\beta_i+\beta_j)+(H^j-H^i) < 0$, and truncate it be $\beta_i+\beta_j$ if $\lambda(\beta_i+\beta_j)+(H^j-H^i) > 2\lambda(\beta_i+\beta_j)$. However, this strategy requires multiple iterations since only a pair of elements in $\{\beta_v\}_{v=1}^m$ can be updated together. To address this issue, we propose a more efficient solution that allows updating all the $\beta_v$ simultaneously, explicitly eliminating the need for iteration. By taking advantage of the coordinate descent method, we can filter out the valid elements that are not governed by the boundary constraints, formally denoting the valid index set as $\mathcal{I}$. Consequently, the inequality constraints of $0 \leq \beta_v \leq 1$ are slack to the weight set $\{\beta_v\}_{v\in\mathcal{I}}$ and the optimization problem can be directly solved. By introducing a Lagrangian multiplier $\eta$, the Lagrangian function $\mathcal{L}(\beta, \eta)$ can be formally defined as:

$$\mathcal{L}(\beta, \eta) = \sum_{v\in\mathcal{I}} \beta_v H^v + \frac{1}{2}\lambda\|\beta\|_2^2 + \eta(1 - \sum_{v\in\mathcal{I}} \beta_v). \tag{A.23}$$

The corresponding Karush-Kuhn-Tucker (KKT) conditions can then be formulated as:

$$\begin{cases} \nabla_{\beta_v}\mathcal{L}(\beta, \eta) = \dfrac{\partial\mathcal{L}(\beta, \eta)}{\partial\beta_v} = H^v + \lambda\beta_v - \eta = 0, \quad v \in \mathcal{I}, \\ \nabla_{\eta}\mathcal{L}(\beta, \eta) = \dfrac{\partial\mathcal{L}(\beta, \eta)}{\partial\eta} = 1 - \displaystyle\sum_{v\in\mathcal{I}} \beta_v = 0. \end{cases} \tag{A.24}$$

Note that we have already taken the equation constraint $\sum_{v\in\mathcal{I}} \beta_v = 1$ into consideration when deriving the representation of $\nabla_{\beta_v}\mathcal{L}(\beta, \eta)$. The optimal result can be obtained by solving the $|\mathcal{I}| + 1$ equations. By summing up all the $\nabla_{\beta_v}\mathcal{L}(\beta, \eta)$ along $v$ within $\mathcal{I}$, the Lagrangian multiplier $\eta$ can be obtained as:

$$\eta = \frac{\sum_{v'\in\mathcal{I}} H^{v'} + \lambda}{|\mathcal{I}|}. \tag{A.25}$$

Therefore, by taking $\eta$ back into the KKT conditions, we can obtain the optimal solution of $\beta_v$, following:

$$\beta_v^* = \frac{\sum_{v'\in\mathcal{I}} H^{v'} - |\mathcal{I}|H^v + \lambda}{\lambda|\mathcal{I}|}, v \in \mathcal{I}. \tag{A.26}$$

Since all the weight $\beta_v$ should satisfy the inequality constraint $0 \leq \beta_v \leq 1$, the above relationships provide an effective strategy to determine the valid index set $\mathcal{I}$, i.e., the corresponding $H^v$ in $\mathcal{I}$ should satisfy $H^v \leq (\sum_{v'\in\mathcal{I}} H^{v'} + \lambda)/|\mathcal{I}|$. Therefore, we can develop a formalize definition of the valid index set, as follows:

$$\mathcal{I} = \big\{v | H^v < (\sum_{v'\in\mathcal{I}} H^{v'} + \lambda)/|\mathcal{I}|, v = 1, 2, \ldots, m\big\}. \tag{A.27}$$

In practical implementation, we first sort all $H^v$ in descending order and then sequentially remove the indices that fail to satisfy the constraint of Eq. (A.27), leading to the valid set $\mathcal{I}$. The optimal result can be obtained in a single round of iteration, with the resulting weight set $\{\beta_v^*\}_{v=1}^m$ given by:

$$\beta_v^* = \begin{cases} \dfrac{\sum_{v'\in\mathcal{I}} H^{v'} - |\mathcal{I}|H^v + \lambda}{\lambda|\mathcal{I}|}, v \in \mathcal{I}, \\ 0, \quad v \in \{1, 2, \ldots, m\}/\mathcal{I}. \end{cases} \tag{A.28}$$

## A.3. Overall Optimization

The overall optimization problem for Bidirectional Collaborative Diffusion can be solved by recursively optimizing $F$ and $\beta$ until convergence following Section A.1 and Section A.2, respectively. Additionally, by leveraging the conjugate gradient method, the iterative update of $F$ achieves a faster convergence rate, as outlined in Algorithm 2. The resulting $F^*$ effectively captures the underlying manifold structure while automatically reducing the adverse effects of inappropriate connections. As a result, it exhibits lower sensitivity to the affinity graph construction strategy and ensures a more robust similarity representation.

## B. Thermodynamic Markovian Transition

**Discrete Wasserstein Distance**  Given two discrete probability mass distributions $\boldsymbol{p}_{\text{start}}, \boldsymbol{p}_{\text{end}} \in \mathbb{R}^n$, a transport problem can be formulated from $\boldsymbol{p}_{\text{start}}$ moving towards $\boldsymbol{p}_{\text{end}}$. Define $\boldsymbol{C} \in \mathbb{R}^{n \times n}$ as the cost matrix, where $\boldsymbol{C}[i,j] \geq 0$ represents the cost required to transport a unit from the $i$-th position $\boldsymbol{p}_{\text{start}}[i]$ to the $j$-th position $\boldsymbol{p}_{\text{end}}[j]$ (also denoted as edge $(i,j)$). The matrix $\boldsymbol{Q} \in \mathbb{R}^{n \times n}$ quantifies the transport strategy, where $\boldsymbol{Q}[i,j]$ indicates the amount being transported from $\boldsymbol{p}_{\text{start}}[i]$ to $\boldsymbol{p}_{\text{end}}[j]$, subject to the following requirements:

$$\boldsymbol{p}_{\text{start}}[i] = \sum_{j=1}^{n} \boldsymbol{Q}[i,j],$$
$$\boldsymbol{p}_{\text{end}}[j] = \sum_{i=1}^{n} \boldsymbol{Q}[i,j]. \tag{B.1}$$

The Wasserstein distance is defined as the smallest cost required to transport one distribution to another, considering all possible transportation matrices:

$$\mathcal{W}_1(\boldsymbol{p}_{\text{start}}, \boldsymbol{p}_{\text{end}}) := \min_{\boldsymbol{Q}} \sum_{i,j} \boldsymbol{Q}[i,j]\boldsymbol{C}[i,j]. \tag{B.2}$$

Indeed, the transport matrix $\boldsymbol{Q}^*$ that yields the optimal solution in Eq. (B.2) represents the optimal transport between the two distributions, under the cost defined by $\boldsymbol{C}$.

**Definition of Thermodynamic Markovian Transition Cost**  Each instance $x_i$ in the set $\chi$ is assigned a probability distribution $\boldsymbol{p}_i \in \mathbb{R}^n$ on $\chi$. For two distinct distributions $\boldsymbol{p}_i$ and $\boldsymbol{p}_j$, we have defined their distance by introducing a thermodynamic Markovian transition between them. We first pick a path $\{\boldsymbol{p}_{i_0}, \boldsymbol{p}_{i_1}, \ldots, \boldsymbol{p}_{i_K}\} = \pi(\{\boldsymbol{p}_i\}_{i=1}^n)$ with each succeeding distribution $\boldsymbol{p}_{i_{k+1}}$ lying within the local region of $\boldsymbol{p}_{i_k}$. Then, a time-dependent transition flow $\boldsymbol{q}_t$ is present to go through the temporary states in the path $\pi$, satisfying the discretized master equation of flow dynamics. More specifically, given $\boldsymbol{p}_i, \boldsymbol{p}_j$ and a path $\pi$, we define the flow $\boldsymbol{q}_t$ to satisfy the following conditions:

$$\begin{aligned} \textbf{(Flow Dynamics)} \quad & \dot{\boldsymbol{q}}_t = \boldsymbol{T}_t \boldsymbol{q}_t, \quad \text{for } t \in [0, \tau], \text{a.e.,} \\ \textbf{(Temperate States)} \quad & \boldsymbol{q}_{\frac{k\tau}{K}} = \boldsymbol{p}_{i_k}, \quad \text{for } k = 0, 1, \ldots, K, \\ \textbf{(Boundary Conditions)} \quad & \boldsymbol{q}_0 = \boldsymbol{p}_{i_0} = \boldsymbol{p}_i, \quad \boldsymbol{q}_\tau = \boldsymbol{p}_{i_K} = \boldsymbol{p}_j, \end{aligned} \tag{B.3}$$

where $\boldsymbol{T}_t$ is a transition rate matrix satisfying the following condition:

$$\boldsymbol{T}_t[r,r] = -\sum_{s \neq r} \boldsymbol{T}_t[s,r], \quad \text{for } r = 1, 2, \ldots, n. \tag{B.4}$$

The entire transition cost $C(\boldsymbol{q}_t)$ for the flow $\boldsymbol{q}_t$ at time $t$ is determined by summing the transition costs across all graph edges, and it can be divided into $K$ stages:

$$C(\boldsymbol{q}_t) = \int_0^\tau \sum_{\substack{r,s=1 \\ r<s}}^n |J(r,s,t)|\, d(r,s)\mathrm{dt} = \sum_{k=0}^{K-1} \int_{\frac{k\tau}{K}}^{\frac{(k+1)\tau}{K}} \sum_{\substack{r,s=1 \\ r<s}}^n |J(r,s,t)|\, d(r,s)\mathrm{dt}, \tag{B.5}$$

where $J(r,s,t)$ is the transition current from the $r$-th position to the $s$-th position at time $t$,

$$J(r,s,t) = \boldsymbol{T}_t[r,s]\boldsymbol{q}_t[s] - \boldsymbol{T}_t[s,r]\boldsymbol{q}_t[r], \tag{B.6}$$

and the Euclidean distance in feature space $d(r, s)$ serves as the cost of this current. The distance between $\boldsymbol{p}_i$ and $\boldsymbol{q}_i$ is defined by taking the minimum of $C(\boldsymbol{q}_t)$ over all feasible path $\pi$ and transition matrix $\boldsymbol{T}_t$:

$$d'(i, j) := \min_{\pi, \boldsymbol{T}_t} C(\boldsymbol{q}_t). \tag{B.7}$$

We have the following equivalency between $d'(i, j)$ and $L^1$-Wasserstein distance $\mathcal{W}_1$:

$$d'(i, j) = \min_{\pi} \sum_{k=0}^{K-1} \mathcal{W}_1(\boldsymbol{p}_{i_k}, \boldsymbol{p}_{i_{k+1}}). \tag{B.8}$$

## B.1. Entropy Production

In this section, we will discuss the relationship between our flow transition cost and the entropy production of stochastic thermodynamics. Briefly, the probability current $J(r, s, t)$ bridges this relationship, inspired by the previous physically based analysis (Van Vu & Saito, 2023; Seifert, 2012; Barato & Seifert, 2015; Ito, 2018).

The entropy production $\Delta S$ during the process consists of two components, the changes in the entropy of the system and the environment,

$$\Delta S = \Delta S_{\text{sys}} + \Delta S_{\text{env}}. \tag{B.9}$$

The system entropy, characterized by the Shannon entropy, can be expressed as

$$S_{\text{sys}}(\boldsymbol{q}_t) = -\sum_{r=1}^{n} \boldsymbol{q}_t[r] \log \boldsymbol{q}_t[r] = S_{\text{sys}}(\boldsymbol{q}_0) + \int_0^{\tau} -\sum_{r=1}^{n} \dot{\boldsymbol{q}}_t[r](\log \boldsymbol{q}_t[r] + 1)\mathrm{d}t. \tag{B.10}$$

And the change of the system entropy is

$$\Delta S_{\text{sys}} = S_{\text{sys}}(\boldsymbol{q}_\tau) - S_{\text{sys}}(\boldsymbol{q}_0) = \int_0^{\tau} -\sum_{r=1}^{n} \dot{\boldsymbol{q}}_t[r](\log \boldsymbol{q}_t[r] + 1)\mathrm{d}t = -\int_0^{\tau} \sum_{r,s=1}^{n} \boldsymbol{T}_t[r, s]\boldsymbol{q}_t[s](\log \boldsymbol{q}_t[r] + 1)\mathrm{d}t. \tag{B.11}$$

By substituting $\boldsymbol{T}_t[r, r]$ using Eq. (B.4), the system entropy change can be further simplified:

$$\begin{aligned}
\Delta S_{\text{sys}} &= -\int_0^{\tau} \sum_{\substack{r,s=1 \\ r\neq s}}^{n} (\boldsymbol{T}_t[r, s]\boldsymbol{q}_t[s] - \boldsymbol{T}_t[s, r]\boldsymbol{q}_t[r])(\log \boldsymbol{q}_t[r] + 1)\mathrm{d}t \\
&= -\int_0^{\tau} \sum_{\substack{r,s=1 \\ r<s}}^{n} (\boldsymbol{T}_t[r, s]\boldsymbol{q}_t[s] - \boldsymbol{T}_t[s, r]\boldsymbol{q}_t[r])\left(\log \frac{\boldsymbol{q}_t[r]}{\boldsymbol{q}_t[s]}\right)\mathrm{d}t.
\end{aligned} \tag{B.12}$$

To compute the system entropy production, we should consider how the transition contributes to the entropy change in the environment. According to Van Vu & Saito (2023), we can assume the transition rates $\boldsymbol{T}_t$ to further satisfy the local detailed balance condition, in the case of microscopically reversible dynamics:

$$\log \frac{\boldsymbol{T}_t[r, s]}{\boldsymbol{T}_t[s, r]} = \boldsymbol{S}_t[r, s], \text{ and } \boldsymbol{T}_t[r, s] > 0 \text{ if and only if } \boldsymbol{T}_t[s, r] > 0, \tag{B.13}$$

where $\boldsymbol{S}_t[r, s]$ denotes the environmental entropy change due to the transition from the $r$-th position to the $s$-th position. The total environment entropy change is obtained by multiplying $\boldsymbol{S}_t[r, s]$ and the amount transferred from position $r$ to $s$,

$$\begin{aligned}
\Delta S_{\text{env}} &= \int_0^{\tau} \sum_{\substack{r,s=1 \\ r\neq s}}^{n} \boldsymbol{T}_t[r, s]\boldsymbol{q}_t[s] \cdot \boldsymbol{S}_t[r, s]\mathrm{d}t \\
&= \int_0^{\tau} \sum_{\substack{r,s=1 \\ r<s}}^{n} (\boldsymbol{T}_t[r, s]\boldsymbol{q}_t[s] - \boldsymbol{T}_t[s, r]\boldsymbol{q}_t[r]) \cdot \boldsymbol{S}_t[r, s]\mathrm{d}t.
\end{aligned} \tag{B.14}$$

Consequently, the entropy production can be obtained:

$$\Delta S = \Delta S_{\text{sys}} + \Delta S_{\text{env}} = \int_0^\tau \sum_{\substack{r,s=1 \\ r<s}}^n |J(r,s,t)| \cdot \left| \log \frac{\boldsymbol{T_t}[r,s]\boldsymbol{q}_t[s]}{\boldsymbol{T_t}[s,r]\boldsymbol{q}_t[r]} \right| \mathrm{d}t. \tag{B.15}$$

Compare the expression of the entropy production Eq. (B.15) and the transition cost Eq. (B.5). Both take the form of a weighted summation over the flow current $J(r,s,t)$. For entropy production, the weighting factor corresponds to the so-called thermodynamic force on the edge $(r,s)$. In contrast, for our transition cost, we replace this factor with the transport cost associated with the edge $(r,s)$.

## B.2. Proof of Eq. (B.8)

For a given path $\pi$, it suffices to demonstrate that at each step of the transition flow within the local region, the transition cost matches the Wasserstein distance:

$$\min_{\boldsymbol{T}_t} \int_{t_k}^{t_{k+1}} \sum_{\substack{r,s=1 \\ r<s}}^n |J(r,s,t)| \, d(r,s)\mathrm{d}t = \mathcal{W}_1(\boldsymbol{p}_{i_k}, \boldsymbol{p}_{i_{k+1}}), \tag{B.16}$$

where we define $t_k = k\tau/K$ for simplicity. It is important to note that the optimization is only concerned with the values of $\boldsymbol{T}_t$ within the interval $[t_k, t_{k+1}]$. Inspired by the physically based results in Van Vu & Saito (2023), we establish this equivalence by proving LHS$\geq$RHS and LHS$\leq$RHS.

**Proof of LHS$\geq$RHS**  The integral on the left in Eq. (B.16) can be expressed using Riemann sums. We partition the time interval $[t_k, t_{k+1}]$ into $M$ equal segments. Defining $\Delta t = (t_{k+1} - t_k)/M$, the expression for the sub-interval $[t_k + m\Delta t, t_k + (m+1)\Delta t]$ in the Riemann sum becomes:

$$\sum_{\substack{r,s=1 \\ r<s}}^n |J(r,s,t_k + m\Delta t)| \, d(r,s)\Delta t, \quad m = 0, 1, \ldots, M-1. \tag{B.17}$$

Consider Eq. (B.17) in the context of transportation. It outlines the movement of an amount $|J(r,s,t_k+m\Delta t)|\,\Delta t$ along the edge $(r,s)$, incurring a cost of $\boldsymbol{C}[r,s] = \boldsymbol{C}[s,r] = d(r,s)$. Additionally, if the transition current $J$ is positive, this mass moves from position $r$ to position $s$. Conversely, if $J$ is negative, the direction is from $s$ to $r$. Consequently, we can construct a matrix $\boldsymbol{Q}$ to represent the transported quantity during the interval $[t_k, t_{k+1}]$, defined as follows:

$$\boldsymbol{Q}[r,s] = \sum_{m=0}^{M-1} [J(r,s,t_k + m\Delta t)]_+ \, \Delta t,$$

$$\boldsymbol{Q}[s,r] = \sum_{m=0}^{M-1} [J(r,s,t_k + m\Delta t)]_- \, \Delta t, \tag{B.18}$$

where $[x]_+ = x$ if $x$ is positive, and vanishes if $x$ is negative, while $[x]_- = |x| - [x]_+$ captures the negative component.

The Riemann sum $S_M$ can thus be expressed using $\boldsymbol{Q}$ and $\boldsymbol{C}$, in connection with the definition of $\mathcal{W}_1$:

$$S_M = \sum_{m=0}^{M-1} \sum_{\substack{r,s=1 \\ r<s}}^n |J(r,s,t_k + m\Delta t)| \, d(r,s)\Delta t = \sum_{r,s=1}^n \boldsymbol{Q}[r,s]\boldsymbol{C}[r,s] \geq \mathcal{W}_1(\boldsymbol{p}_{i_k}, \boldsymbol{p}_{i_{k+1}}) = \text{RHS}. \tag{B.19}$$

As $M \to \infty$ and $\Delta t \to 0$, the Riemann sum $S_M$ approaches the integral on the left-hand side of Eq. (B.16), resulting in LHS$\geq$RHS.

**Proof of LHS$\leq$RHS**  Given a matrix $\boldsymbol{Q}$ that optimally determines the transportation amount, solving $\mathcal{W}_1(\boldsymbol{p}_{i_k}, \boldsymbol{p}_{i_{k+1}})$, our task is to formulate a valid $\boldsymbol{T}_t$ for the flow $\boldsymbol{q}_t$ over the interval $[t_k, t_{k+1}]$ to also attain this value. According to the definition, every element $\boldsymbol{Q}[r,s]$ indicates the total amount moved from the $r$-th to the $s$-th position. We can break down the optimal transport plan into a series of $M'$ vertex-to-vertex transfers, each occurring over a time span of $\Delta t' = (t_{k+1} - t_k)/M'$. Specifically, the transport within $[t_k, t_{k+1}]$ can be described as follows:

- Within $[t_k, t_k + \Delta t']$, transport $q_0$ from $r_0$ to $s_0$;

- Within $[t_k + \Delta t', t_k + 2\Delta t']$, transport $q_1$ from $r_1$ to $s_1$;

- ...

- Within $[t_k + (M' - 1)\Delta t', t_k + M'\Delta t']$, transport $q_{M'}$ from $r_{M'-1}$ to $s_{M'-1}$;

For each step $0 \le m \le M' - 1$, the transported quantity $q_m > 0$ must not exceed the available mass at position $r_k$. The optimal Wasserstein distance can be expressed in terms of $q_m$ and the pairs $(r_m, s_m)$ by summing them up:

$$\mathcal{W}_1(\boldsymbol{p}_{i_k}, \boldsymbol{p}_{i_{k+1}}) = \sum_{m=0}^{M'-1} q_m d(r_m, s_m). \tag{B.20}$$

We can now define a transition flow $\boldsymbol{q}_t$ for $t \in [t_k, t_{k+1}]$ using the decomposition approach: initialize at $\boldsymbol{q}_{t_k} = \boldsymbol{p}_{i_k}$. During the $m$-th sub-interval $[t_k + m\Delta t', t_k + (m+1)\Delta t']$, the flow evolves linearly,

$$
\begin{aligned}
\boldsymbol{q}_t[r_m] &= \boldsymbol{q}_{t_k+m\Delta t'}[r_m] - \frac{t - t_k - m\Delta t'}{\Delta t'} q_m, \\
\boldsymbol{q}_t[s_m] &= \boldsymbol{q}_{t_k+m\Delta t'}[s_m] + \frac{t - t_k - m\Delta t'}{\Delta t'} q_m, \\
\boldsymbol{q}_t[r] &= \boldsymbol{q}_{t_k+m\Delta t'}[r], \quad \text{for } r \ne r_m, s_m.
\end{aligned}
\tag{B.21}
$$

The corresponding time-derivative is $\dot{\boldsymbol{q}}_t$, where all channels are zeros except at $r_m$ and $r_s$,

$$\dot{\boldsymbol{q}}_t[r_m] = -q_m/\Delta t', \quad \dot{\boldsymbol{q}}_t[s_m] = q_m/\Delta t'. \tag{B.22}$$

Hence, we can obtain the transition matrix $\boldsymbol{T}_t$ derived from $\boldsymbol{q}_t$, where most entries are zero except for four specific channels: $[r_m, r_m]$, $[r_m, s_m]$, $[s_m, s_m]$, and $[s_m, r_m]$. These channels comply with the following equations, dictated by the discrete master equations:

$$
\begin{aligned}
\boldsymbol{T}_t[r_m, r_m] &= -\boldsymbol{T}_t[s_m, r_m], \\
\boldsymbol{T}_t[s_m, s_m] &= -\boldsymbol{T}_t[r_m, s_m], \\
\dot{\boldsymbol{q}}_t[r_m] &= \boldsymbol{T}_t[r_m, r_m]\boldsymbol{q}_t[r_m] + \boldsymbol{T}_t[r_m, s_m]\boldsymbol{q}_t[s_m], \\
\dot{\boldsymbol{q}}_t[s_m] &= \boldsymbol{T}_t[s_m, r_m]\boldsymbol{q}_t[r_m] + \boldsymbol{T}_t[s_m, s_m]\boldsymbol{q}_t[s_m].
\end{aligned}
\tag{B.23}
$$

Through straightforward calculations, it is found that any $\boldsymbol{T}_t[r_m, s_m]$ and $\boldsymbol{T}_t[s_m, r_m]$ satisfying the following condition:

$$\boldsymbol{T}_t[r_m, s_m]\boldsymbol{q}_t[s_m] - \boldsymbol{T}_t[s_m, r_m]\boldsymbol{q}_t[r_m] = -q_m/\Delta t' \tag{B.24}$$

constitutes the solution. By introducing a parameter $\theta > 0$, we can guarantee the existence,

$$\boldsymbol{T}_t[r_m, s_m] = \frac{\theta q_m}{\boldsymbol{q}_t[s_m]\Delta t'}, \quad \boldsymbol{T}_t[s_m, r_m] = \frac{(\theta + 1)q_m}{\boldsymbol{q}_t[r_m]\Delta t'}. \tag{B.25}$$

Now that the transition matrix $\boldsymbol{T}_t$ is determined. We can compute the transition cost in the interval $[t_k, t_{k+1}]$ by decomposing the integral into the $M'$ stages:

$$\int_{t_k}^{t_{k+1}} \sum_{\substack{r,s=1 \\ r<s}}^{n} |J(r, s, t)|\, d(r, s)\mathrm{dt} = \sum_{m=0}^{M'-1} \int_{t_k+m\Delta t'}^{t_k+(m+1)\Delta t'} \sum_{\substack{r,s=1 \\ r<s}}^{n} |J(r, s, t)|\, d(r, s)\mathrm{dt}. \tag{B.26}$$

At each stage, the current $J(r, s, t)$ vanishes unless it is the current on the edge $(r_m, s_m)$. Hence, we can simplify the integral as:

$$\int_{t_k+m\Delta t'}^{t_k+(m+1)\Delta t'} \sum_{\substack{r,s=1 \\ r<s}}^{n} |J(r, s, t)|\, d(r, s)\mathrm{dt} = \int_{t_k+m\Delta t'}^{t_k+(m+1)\Delta t'} |J(r_m, s_m, t)|\, d(r_m, s_m)\mathrm{dt}. \tag{B.27}$$

Substitute the results in Eq. (B.24) into the definition of $J$ to obtain that

$$|J(r_m, s_m, t)| = |\boldsymbol{T}_t[r_m, s_m]\boldsymbol{q}_t[s_m] - \boldsymbol{T}_t[s_m, r_m]\boldsymbol{q}_t[r_m]| = q_m/\Delta t'. \tag{B.28}$$

This results in the simplification of transition cost as:

$$\int_{t_k}^{t_{k+1}} \sum_{\substack{r,s=1 \\ r<s}}^{n} |J(r, s, t)|\, d(r, s)\mathrm{d}t = \sum_{m=0}^{M'-1} \int_{t_k+m\Delta t'}^{t_k+(m+1)\Delta t'} \frac{q_m}{\Delta t'} \cdot d(r_m, s_m)\mathrm{d}t = \sum_{m=0}^{M'-1} q_m d(r_m, s_m). \tag{B.29}$$

Compare the result with Eq. (B.20), we have construct a transition flow $\boldsymbol{q}_t$ that definitely achieves the Wasserstein distance $\mathcal{W}_1(\boldsymbol{p}_{i_k}, \boldsymbol{p}_{i_{k+1}})$. Consequently, we have proved that LHS$\leq$RHS in Eq. (B.16).

## B.3. Iterative solution for $\mathcal{W}_1$

At this point, the complicated task of resolving the thermodynamic energy in Eq. (B.7) has been reformulated into the optimal transport problem in Eq. (B.8). To efficiently solve this optimal transport problem and determine $\mathcal{W}_1$, we incorporate an additional entropy regularization term. Given that the two distributions involved are $\boldsymbol{p}_{\text{start}}$ and $\boldsymbol{p}_{\text{end}}$, this regularization is introduced as follows:

$$\mathcal{W}_{1,\varepsilon}(\boldsymbol{p}_{\text{start}}, \boldsymbol{p}_{\text{end}}) := \min_{\boldsymbol{Q}} \sum_{i,j=1}^{n} \boldsymbol{Q}[i,j]\boldsymbol{C}[i,j] - \varepsilon H(\boldsymbol{Q}), \tag{B.30}$$

where the conditions in Eq. (B.1) are also required. Following (Cuturi, 2013; Cuturi & Doucet, 2014), the following entropy regularization term is incorporated,

$$-H(\boldsymbol{Q}) = \sum_{i,j=1}^{n} \boldsymbol{Q}[i,j] \log \boldsymbol{Q}[i,j] - \boldsymbol{Q}[i,j]. \tag{B.31}$$

We can obtain with the Lagrange function $\mathcal{L}(\boldsymbol{Q}, \boldsymbol{f}, \boldsymbol{g})$:

$$\mathcal{L}(\boldsymbol{Q}, \boldsymbol{f}, \boldsymbol{g}) = \sum_{i,j=1}^{n} \boldsymbol{Q}[i,j]\boldsymbol{C}[i,j] - \varepsilon H(\boldsymbol{Q}) - \sum_{i=1}^{n} \boldsymbol{f}[i]\left(\sum_{j=1}^{n} \boldsymbol{Q}[i,j] - \boldsymbol{p}_{\text{start}}[i]\right) - \sum_{j=1}^{n} \boldsymbol{g}[j]\left(\sum_{i=1}^{n} \boldsymbol{Q}[i,j] - \boldsymbol{p}_{\text{end}}[j]\right), \tag{B.32}$$

where $\boldsymbol{f}$ and $\boldsymbol{g}$ are Lagrangian multipliers. By taking the derivative with respect to $\boldsymbol{Q}$, we obtain the first-order condition

$$\frac{\mathcal{L}(\boldsymbol{Q}, \boldsymbol{f}, \boldsymbol{g})}{\partial \boldsymbol{Q}[i,j]} = \boldsymbol{C}[i,j] + \varepsilon \log \boldsymbol{Q}[i,j] - \boldsymbol{f}[i] - \boldsymbol{g}[i] = 0, \tag{B.33}$$

and the solution

$$\boldsymbol{Q} = \text{diag}(e^{\boldsymbol{f}/\varepsilon}) \cdot e^{-\boldsymbol{C}/\varepsilon} \cdot \text{diag}(e^{\boldsymbol{g}/\varepsilon}). \tag{B.34}$$

Here, the operation $\text{diag}(\cdot)$ takes the input vector to a diagonal matrix. The expression for $\boldsymbol{Q}$ can be determined using a fixed-point iteration method as outlined below:

(1) Begin by initializing $\boldsymbol{u}^{(0)}$ and $\boldsymbol{v}^{(0)}$ as $n$-dimensional vectors where each element is set to 1. Define $\boldsymbol{K} = e^{-\boldsymbol{C}/\varepsilon}$ such that $\boldsymbol{K}[i,j] = e^{-\boldsymbol{C}[i,j]/\varepsilon}$.

(2) In each iteration, perform the following computations:

$$\begin{aligned}
\boldsymbol{u}^{(t+1)} &= \boldsymbol{p}_{\text{start}} \oslash \left(\boldsymbol{K} \cdot \boldsymbol{v}^{(t)}\right), \\
\boldsymbol{v}^{(t+1)} &= \boldsymbol{p}_{\text{end}} \oslash \left(\boldsymbol{K}^\top \cdot \boldsymbol{u}^{(t+1)}\right),
\end{aligned} \tag{B.35}$$

and continue until convergence, which results in the pair $(\boldsymbol{u}^*, \boldsymbol{v}^*)$. The Wasserstein distance $\mathcal{W}_{1,\varepsilon}$ can be then derived after the optimal matrix $\boldsymbol{Q} = \text{diag}(\boldsymbol{u}^*) \cdot \boldsymbol{K} \cdot \text{diag}(\boldsymbol{v}^*)$ is determined. Such that the overall time complexity is $\mathcal{O}(n^3)$.

# C. Experiments

## C.1. Visualization

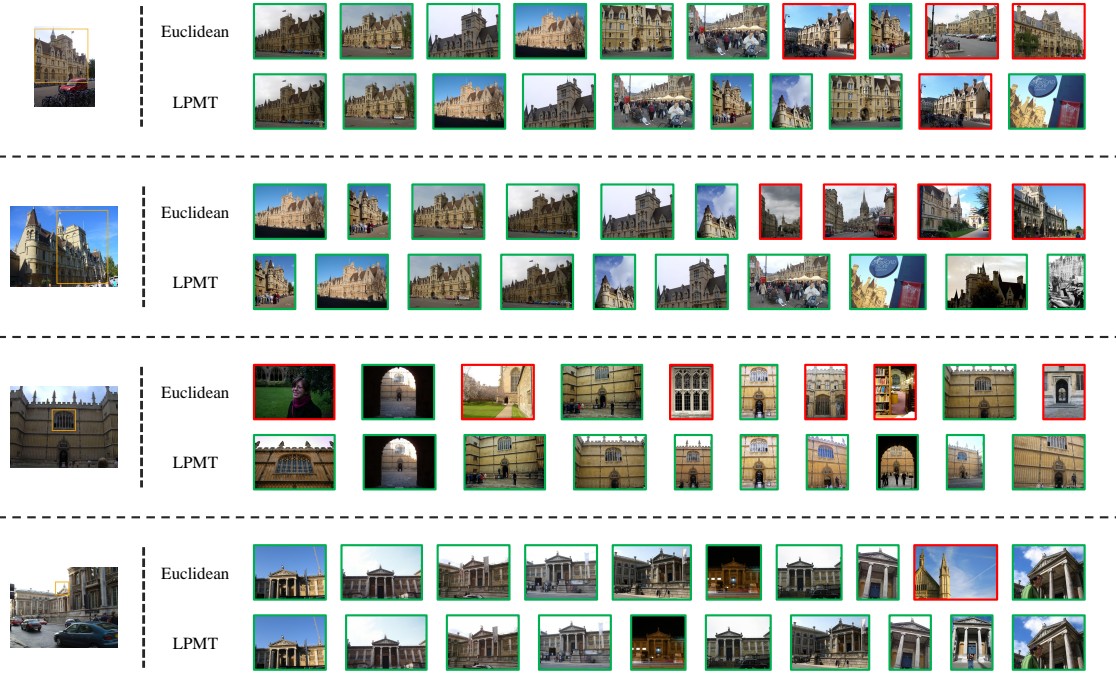

*Figure 6.* This figure showcases the quantitative evaluation of our proposed LPMT in comparison with retrieval results based on Euclidean distance. The region of interest in the query image is highlighted by an orange bounding box on the left. On the right, we visualize the retrieval performance of LPMT against Euclidean distance by displaying the top 10 ranked images for both approaches. Correct matches (true positives) are enclosed in green bounding boxes, whereas incorrect ones (false matches) are marked in red, demonstrating the effectiveness of LPMT as an accurate and reliable distance metric for instance retrieval.

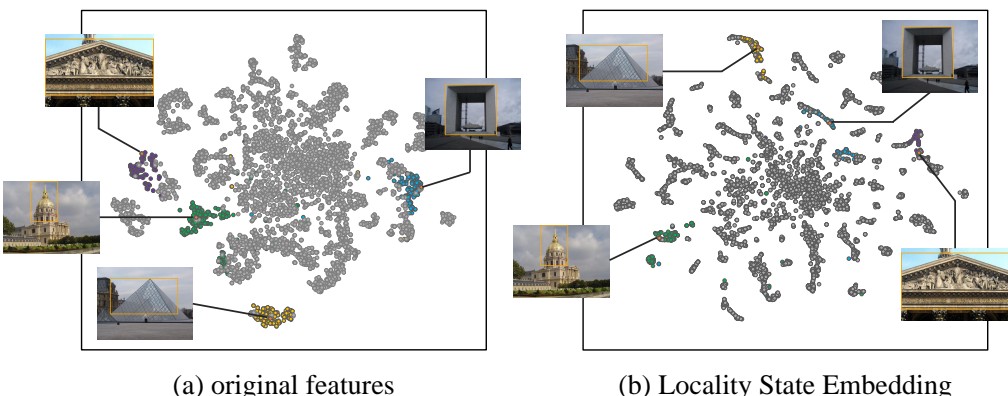

(a) original features          (b) Locality State Embedding

*Figure 7.* (a) The t-SNE visualization of the original image features directly extracted by the deep retrieval model. (b) The t-SNE visualization of the distributions produced by the Locality State Embedding (LSE) strategy using cosine similarity weights. Compared to the original image features, LSE helps mitigate the influence of outliers and leads to improved clustering and retrieval performance.

## C.2. Extended Results

*Table 7.* Evaluation of the retrieval performances based on global image features extracted by MAC (Tolias et al., 2016).

| Method | Easy | | Medium | | Hard | |
|---|---|---|---|---|---|---|
| | *R*Oxf | *R*Par | *R*Oxf | *R*Par | *R*Oxf | *R*Par |
| MAC | 47.2 | 69.7 | 34.6 | 55.7 | 14.3 | 32.6 |
| AQE | 54.4 | 80.9 | 40.6 | 67.0 | 17.1 | 45.2 |
| $\alpha$QE | 50.3 | 77.8 | 37.1 | 64.4 | 16.3 | 43.0 |
| SG | 46.1 | 75.9 | 36.1 | 60.4 | 16.6 | 38.8 |
| STML | 61.4 | 86.8 | 46.7 | 76.9 | 22.3 | 59.5 |
| AQEwD | 52.8 | 79.6 | 39.7 | 65.0 | 17.3 | 42.9 |
| DFS | 54.6 | 83.8 | 40.6 | 74.0 | 18.8 | 58.1 |
| RDP | 59.0 | 85.2 | 45.3 | 76.3 | 21.4 | 58.9 |
| CAS | 68.6 | 90.1 | 52.9 | 82.3 | 30.4 | 68.1 |
| GSS | 60.0 | 87.5 | 45.4 | 76.7 | 22.8 | 59.7 |
| ConAff | 65.5 | 88.7 | 50.1 | 79.3 | 25.6 | 62.4 |
| **LPMT** | **71.0** | **91.2** | **53.6** | **83.5** | **31.2** | **69.3** |

*Table 8.* Evaluation of the retrieval performances based on global image features extracted by R-MAC (Tolias et al., 2016).

| Method | Easy | | Medium | | Hard | |
|---|---|---|---|---|---|---|
| | *R*Oxf | *R*Par | *R*Oxf | *R*Par | *R*Oxf | *R*Par |
| R-MAC | 61.2 | 79.3 | 40.2 | 63.8 | 10.1 | 38.2 |
| AQE | 69.4 | 85.7 | 47.8 | 71.1 | 15.9 | 47.9 |
| $\alpha$QE | 64.9 | 84.7 | 42.8 | 70.8 | 11.4 | 47.8 |
| SG | 60.1 | 84.9 | 42.7 | 68.4 | 16.5 | 45.4 |
| STML | 71.8 | 88.7 | 53.2 | 78.2 | 23.4 | 58.8 |
| AQEwD | 70.5 | 85.9 | 48.7 | 70.7 | 15.3 | 46.9 |
| DFS | 70.0 | 87.5 | 51.8 | 78.8 | 20.3 | 63.5 |
| RDP | 73.7 | 88.8 | 54.3 | 79.6 | 22.2 | 61.3 |
| CAS | 82.6 | 90.0 | 62.5 | 82.5 | 34.1 | 67.4 |
| GSS | 75.0 | 89.9 | 54.7 | 78.5 | 24.4 | 60.5 |
| ConAff | 77.6 | 88.0 | 56.4 | 80.0 | 27.5 | 61.3 |
| **LPMT** | **83.9** | **90.6** | **62.9** | **83.3** | **36.0** | **68.7** |

*Table 9.* Evaluation of the retrieval performances based on global image features extracted by DELG (Cao et al., 2020).

| Method | Easy | | Medium | | Hard | |
|---|---|---|---|---|---|---|
| | *R*Oxf | *R*Par | *R*Oxf | *R*Par | *R*Oxf | *R*Par |
| DELG | 91.0 | 95.1 | 77.4 | 88.2 | 57.5 | 75.2 |
| AQE | 96.1 | 95.1 | 82.6 | 90.2 | 61.3 | 79.5 |
| $\alpha$QE | 94.5 | 96.0 | 81.5 | 90.7 | 63.9 | 80.8 |
| SG | 95.5 | 95.7 | 83.3 | 90.0 | 66.9 | 79.6 |
| STML | 93.3 | 95.1 | 80.3 | 88.2 | 62.1 | 75.3 |
| AQEwD | 96.0 | 96.4 | 83.3 | 90.9 | 66.0 | 80.7 |
| DFS | 87.5 | 93.4 | 74.1 | 88.6 | 48.1 | 77.9 |
| RDP | 94.4 | 95.0 | 84.7 | 91.6 | 66.3 | 81.8 |
| CAS | 97.6 | 94.7 | 87.0 | 91.6 | 72.1 | 82.6 |
| GSS | 97.3 | 95.7 | 84.7 | 90.9 | 66.4 | 81.6 |
| ConAff | 95.5 | 92.4 | 84.2 | 89.4 | 66.7 | 76.3 |
| **LPMT** | **99.2** | **96.4** | **88.6** | **94.0** | **73.7** | **87.1** |

*Table 10.* Evaluation of the retrieval performances based on global image features extracted by SENet (Lee et al., 2023).

| Method | Easy | | Medium | | Hard | |
|---|---|---|---|---|---|---|
| | *R*Oxf | *R*Par | *R*Oxf | *R*Par | *R*Oxf | *R*Par |
| SENet | 94.4 | 94.7 | 81.9 | 90.0 | 63.0 | 78.1 |
| AQE | 95.1 | 95.8 | 82.9 | 92.3 | 65.2 | 82.9 |
| $\alpha$QE | 96.0 | 95.8 | 84.1 | 92.6 | 67.7 | 84.0 |
| SG | 96.5 | 96.2 | 85.5 | 92.1 | 70.3 | 83.1 |
| STML | 95.7 | 95.1 | 84.1 | 89.9 | 67.1 | 77.9 |
| AQEwD | 95.4 | 96.4 | 84.5 | 92.7 | 68.1 | 83.9 |
| DFS | 82.0 | 93.9 | 72.1 | 90.4 | 53.1 | 81.1 |
| RDP | 94.2 | 94.8 | 86.8 | 93.0 | 72.5 | 84.8 |
| CAS | 94.9 | 94.8 | 87.3 | 93.6 | 74.0 | 86.4 |
| GSS | 96.4 | 95.9 | 86.5 | 91.0 | 71.6 | 82.8 |
| ConAff | 96.1 | 93.2 | 87.7 | 92.1 | 74.0 | 81.5 |
| **LPMT** | **96.5** | **96.5** | **89.2** | **95.0** | **76.9** | **88.8** |

