# OpenReview forum: "Locality Preserving Markovian Transition for Instance Retrieval"
_ICML.cc/2025/Conference — ICML 2025 poster_

### Official Review · Reviewer_4j2p · 2025-03-08

**Overall Recommendation:** 4

**Summary:**

This paper tackles the problem of instance retrieval, or finding the image most similar in a dataset to a query image. Existing methods suffer from long-range propagation of similarity information, which the authors improve on with three components. BCD- They improve similarity propagation by combining multiple adjacency graphs rather than relying on a single all-to-all graph.  LSE- transforms distances between pairs into probability distribution between a set of k nearest elements. TMT- they define a new cost metric as the transition cost between 2 elements, which is the term they minimize. They show impressive results across a range of datasets and against a large number of baselines.

**Claims And Evidence:**

Yes, the authors provide extensive and impressive quantitative results against existing works.

**Essential References Not Discussed:**

The works they include in the paper seem reasonable and comprehensive.

**Experimental Designs Or Analyses:**

I am unfamiliar with the current state of the art in instance retrieval, but it all looked reasonable.

**Methods And Evaluation Criteria:**

Yes, the authors provide a comprehensive list of benchmarks and evaluate their proposed method on a set of datasets.

**Other Comments Or Suggestions:**

This paper is very well laid out and easy to follow.

**Other Strengths And Weaknesses:**

This is only a slight weakness, but the task of instance retrieval has limits as far as applicability to real-world scenarios. I can see this work being an interesting basis for future work.

**Questions For Authors:**

I am impressed with the paper presented. I have no recommendations for rebuttal.

**Relation To Broader Scientific Literature:**

While they apply their idea to instance retrieval, I would be very curious to see if their idea can be applied to the attention operation in transformers. A transformer's similarity matrix might be sensitive to exact similarity values between pairs, but a more informed similarity could improve performance. This would increase computational complexity but, depending on the application, could be a worthwhile tradeoff. This could hypothetically be interesting follow-up work.

**Theoretical Claims:**

The methods proposed in this paper are well established in other areas. They borrow ideas from graph theory, probabilistic methods, and optimal transport. When applied to this setting, they all seem fitting.

---

> ### Author Rebuttal · Authors · 2025-04-01
>
> We sincerely appreciate your positive assessment and recognition of our contribution to manifold reranking. In this work, we address the fundamental challenge of manifold ranking by introducing the proposed **Locality Preserving Markovian Transition (LPMT)**. Our approach establishes a structured, long-term transition process that effectively connects distinct distributions within the graph, ensuring reliable information propagation at each step. To mitigate the impact of unreliable connections, BCD adaptively ensembles diffusion processes across multi-level graphs to generate a robust similarity matrix. This is achieved through a joint optimization framework that refines both the combination weights and the diffusion objective, which we solve efficiently using a fixed-point iterative approach. Subsequently, LSE embeds each instance as a probability distribution within the manifold space, and LPMT bridges distant distributions through a sequence of locally constrained transitions. This design not only preserves the intrinsic manifold structure but also maintains essential local characteristics, with the minimal transition cost serving as a principled metric for enhanced retrieval performance.
>
> Building on this foundation, **migrating the manifold ranking algorithm to Transformers presents an intriguing research direction.** As the core of the Transformer architecture, the self-attention mechanism models pairwise token relationships via dot-product similarity between query and key vectors. While stacking Transformer layers facilitates deeper integration of information, it does not inherently capture the underlying manifold structure of token embeddings within each layer. Given the extensive adoption of Transformers, incorporating manifold ranking, despite its additional computational cost, could improve the reliability of similarity estimation. Since our current algorithm is designed for complex manifold spaces and deep learning-based retrieval tasks, further simplifications are necessary to enhance its robustness for broader applications. We will further investigate these aspects in the future.
>
> Regarding **practical deployment**, while our primary focus is on improving manifold ranking effectiveness, several optimizations can enhance efficiency. For computationally intensive operations such as diffusion, an **offline strategy** can be employed to precompute and store results in a database, allowing the LSE distribution of each instance to be maintained in advance. Upon receiving a new query, its probability distribution can be efficiently estimated via linear aggregation of neighboring samples, significantly reducing online computation. Likewise, the exact computation of TMT cost can also be approximated to improve efficiency. By decoupling complex operations into **offline preprocessing and efficient online retrieval**, we can partition the main workload accordingly, making real-world deployment more practical. We will continue investigating these optimizations in future work.
>
> In conclusion, he proposed LPMT framework serves as a strong foundation for effective manifold ranking, demonstrating high performance across multiple retrieval benchmarks. Future work will focus on integrating it with Transformer architectures and optimizing efficiency for practical deployment.

---

### Official Review · Reviewer_3HVn · 2025-03-15

**Overall Recommendation:** 3

**Summary:**

The paper introduces the Locality Preserving Markovian Transition (LPMT) framework to improve instance retrieval by overcoming the limitations of traditional diffusion-based re-ranking methods. Standard methods suffer from diminishing positive signals over long diffusion paths, which weakens their discriminative power. LPMT addresses this by combining Bidirectional Collaborative Diffusion to build robust similarity matrices, Locality State Embedding to encode instances as probability distributions for enhanced local consistency, and a Thermodynamic Markovian Transition process that bridges distant instances via local intermediate states. This integrated approach effectively preserves local relationships while ensuring accurate global retrieval, leading to notable improvements in performance on benchmark datasets.

**Claims And Evidence:**

The paper claims a novel image retrieval strategy based on extracted image features. It constructs an improved similarity matrix using Collaborative Diffusion (BCD), which automatically integrates diffusion processes on multi-level affinity graphs. To enhance global discriminative power without sacrificing local effectiveness, the authors introduce the Locality State Embedding (LSE) strategy, representing each instance as a locally consistent distribution within the underlying manifold. Finally, Thermodynamic Markovian Transition (TMT) is proposed to perform a constrained time evolution process within local regions at each stage. The definition and formulation is easy to follow.

**Essential References Not Discussed:**

No

**Experimental Designs Or Analyses:**

Please check evaluation part comment.

**Methods And Evaluation Criteria:**

Yes, the paper conducts large-scale evaluations on standard datasets, such as Oxford5k (ROxf) and Paris6k, using the standard retrieval metric (mAP), and further divided into Easy, Medium and Hard categories.  I have concerns regarding the setting where an extra collection of one million distractor images is incorporated to form the large-scale ROxf+1M and RPar+1M datasets. Will this operation introduce an inductive bias toward positive results?

**Other Comments Or Suggestions:**

I would like the author to add a pipeline figure to visualize the pipeline in future versions.

**Other Strengths And Weaknesses:**

This method is based on extracted features by off-the-shell models, which could be a limitation for downstream methods though sound theoretical framework has been proposed in this paper. Robustness of this method to weaker features or ablations might be a strong proof of the effectiveness of this framework.

**Questions For Authors:**

1. I have concerns regarding the setting where an extra collection of one million distractor images is incorporated to form the large-scale ROxf+1M and RPar+1M datasets. Will this operation introduce an inductive bias toward positive results?

2. I also have concerns about the time cost associated with building the initial similarity diffusion, as well as whether additional computational optimization is required for incoming data. This could potentially affect the practical application value of this work.

**Relation To Broader Scientific Literature:**

No

**Theoretical Claims:**

Lemma A.1 to Lemma A.5 are well-known results in matrix analysis, how does they relate to Section A.2? Would the assumption of "each
transition only takes place in local regions" be too strong to be generalized?  I would like the authors and other reviewers to double check proof B.2, especially B.21 and B.24 which are not convincing to me.

---

> ### Author Rebuttal · Authors · 2025-04-01
>
> **Q1: Will the incorporation of extra distractor images introduce an inductive bias?**
>
> An additional one million distractor images are introduced to simulate a large-scale image database. Compared to the original ROxf and RPar datasets, the expanded database includes a larger number of hard negative samples for each query, significantly increasing retrieval difficulty (ROxf-M baseline drops from 67.3 to 49.5). When applying reranking methods, these distractors introduce noise into the manifold structure, disrupting information propagation and posing a greater challenge to the algorithm.
>
> **Q2: The role of Lemma A.1 and A.5 in the proof.**
>
> We explicitly present Lemma A.1 and A.5 because they are essential to establish the convexity of the objective function and the convergence of the iterative formulation. In Section A.2, when constraining $\beta$ and updating $F$, we reformulate the objective function as Eq. (A.8) using properties of the Kronecker product. The convexity proof hinges on demonstrating that the Hessian matrix is positive-definite, which is a nontrivial step that requires Lemma A.1 to derive the nature the matrix $S$. Lemma A.5 is instrumental in proving the convergence of the iterative process in Eq. (A.14) and establishing its equivalence to the closed-form solution. Specifically, in Eq. (A.16), the second summation follows the structure of a Neumann series, necessitating the result of Lemma A.5.
>
> While these steps may be derivable for experts in matrix analysis, we include these lemmas to ensure a rigorous and self-contained exposition.
>
> **Q3: Question about the assumption of local transition.**
>
> Limiting each transition to a local region aims to mitigate information loss over multiple iterations in the diffusion process, ensuring reliability at every step. Experiments on various feature databases have demonstrated its effectiveness. We believe that maintaining local characteristics in complex manifold spaces contributes to better overall performance.
>
> **Q4: Explanation of proof B.2.**
>
> We clarify Proof B.2 by summarizing its main idea and then explaining the derivation of Eqs. (B.21) and (B.24).
>
> (1) The core idea is to prove Eq. (B.16) by establishing both directions: RHS ≤ LHS and LHS ≤ RHS. For RHS ≤ LHS, we show that the transition cost of any flow $T_t$ is no less than the Wasserstein distance (We bridge this by introducing a Riemann sum, which is also in the form of a transportation cost). For LHS ≤ RHS, we aim to construct a flow $T_t$ whose cost matches that of a given transport plan $Q$. Note that many such flows may exist; we only need to show existence.
>
> (2) To prove LHS ≤ RHS, we construct $T_t$ through the following steps:
> $Q$ → $(q _ m)$ (B.20) → $q_t$ (B.21) → $T_t$ (B.22–25). In (B.21), we construct a flow $q_t$ with constant velocity for each vertex-to-vertex transport. That’s why we use a linear time parameter $t$ at the vertices $r_m$ and $s_m$, while keeping the other vertices fixed. Eq. (B.24) follows directly from substituting the first and second equations in (B.23) into the third. By replacing $\dot{q}_t$ on the left-hand side with the expression in (B.22), and substituting $T_t[r_m, r_m]$ with $-T_t[s_m, r_m]$, we obtain that $T_t$ satisfies (B.24).
>
> We emphasize again that **we only need to prove the existence** of a valid $T_t$ via construction, uniqueness is not required.
>
> **Q5: Robustness of the proposed method.**
>
> To verify the robustness of our algorithm across different feature databases, we employ not only the classic R-GeM for feature extraction but also more advanced models such as DOLG/CVNet and weaker models like MAC/R-MAC. This enables evaluation across both stronger and weaker features, with our method consistently outperforming these benchmarks (see Appendix C). Additionally, our algorithm also demonstrates reliability in text-image retrieval tasks, as discussed in our response to reviewer ij2o.
>
> **Q6: Time cost and computational optimizations.**
>
> Thank you for raising this important concern. While our primary focus is on improving the effectiveness of manifold ranking, we have some feasible optimizations to address the essential problem of practical deployment.
>
> The BCD component incorporates a diffusion process that operates with a time complexity of $O(n^3)$. Empirically, for a graph with 5000 nodes, executing 12 iterations requires approximately 0.3s. To enhance computational efficiency, we can adopt an offline strategy to precompute high-complexity operations within the database, wherein each image is encoded with a diffusion-based similarity in advance. Upon receiving a query, its diffusion-based similarity can be approximated by linearly aggregating neighboring samples. Similarly, the TMT distance can be precomputed within the graph, enabling reranking with efficiency comparable to kNN. While approximation introduces some performance overhead, we will continue exploring the trade-off between accuracy and efficiency.

---

### Official Review · Reviewer_ij2o · 2025-03-19

**Overall Recommendation:** 3

**Summary:**

Existing re-ranking methods tend to reduce discriminative power over several steps. This paper proposes the LPMT framework for accurate manifold distance measurement, thereby enhancing the retrieval process. The proposed method is supported by several theoretical analyses, and experiments demonstrate significant performance improvements over baseline methods.

**Claims And Evidence:**

The proposed research problem lacks clear articulation and would benefit from more precise formulation. The writing throughout the manuscript requires improvement to enhance readability and comprehension. Regarding empirical validation, the experimental section needs strengthening with more rigorous evaluation protocols and comprehensive analysis of results.

**Essential References Not Discussed:**

The discussion on retrieval methods for textual data requires further expansion.

[1] Su, H., Yen, H., Xia, M., Shi, W ., Muennighoff, N., Wang, H. Y., ... & Yu, T. (2024). Bright: A realistic and challenging benchmark for reasoning-intensive retrieval. arXiv preprint arXiv:2407.12883.
[2] Chen, J., Lin, H., Han, X., & Sun, L. (2024, March). Benchmarking large language models in retrieval-augmented generation. In Proceedings of the AAAI Conference on Artificial Intelligence (Vol. 38, No. 16, pp. 17754-17762).

**Experimental Designs Or Analyses:**

Yes. The experimental design appears comprehensive and provides reasonable support for the work’s central claims.

**Methods And Evaluation Criteria:**

The proposed methods appear to address the stated problem effectively. However, the evaluation criteria require further explanation, as several metrics are not clearly defined or justified in the current presentation.

**Other Comments Or Suggestions:**

It likewise appears to be a weakness.

**Other Strengths And Weaknesses:**

Strengths：

1. The research problem addressed in this paper is significant, as retrieval technology represents an important area of investigation across multiple fields.
2. The experimental results demonstrate performance improvements over the selected baseline methods.

Weakness：

1. The stated claim could be made clearer. For instance, the authors note that instance retrieval focuses on identifying images visually similar to a given query image at a large scale. However, retrieval methods have already been widely used in Large Language Model (LLM) contexts, such as Retrieval-Augmented Generation (RAG). Therefore, it would be beneficial to clarify why the work specifically focuses on image-only retrieval, and whether or how lessons from broader retrieval applications (e.g., text-based retrieval) might be incorporated or contrasted.

2. The motivation for focusing on image retrieval should be more compelling. Highlighting the unique challenges of image retrieval—such as differences from natural language processing (NLP) retrieval tasks—would emphasize the distinctiveness and importance of this work.

3. The datasets used in the experiments appear to be outdated. Recent benchmarks have been introduced for retrieval tasks, including:
[1] Su, H., Yen, H., Xia, M., Shi, W ., Muennighoff, N., Wang, H. Y., ... & Yu, T. (2024). Bright: A realistic and challenging benchmark for reasoning-intensive retrieval. arXiv preprint arXiv:2407.12883.
[2] Chen, J., Lin, H., Han, X., & Sun, L. (2024, March). Benchmarking large language models in retrieval-augmented generation. In Proceedings of the AAAI Conference on Artificial Intelligence (Vol. 38, No. 16, pp. 17754-17762).
Including or comparing against these newer benchmarks could strengthen the evaluation and demonstrate the method’s relevance to current retrieval challenges.

4. The discussion of retrieval-related research is not sufficiently comprehensive, especially regarding developments in the NLP domain. Incorporating relevant NLP-focused retrieval studies and explaining how they align or differ from this work would offer a clearer picture of the broader retrieval landscape.

**Questions For Authors:**

It likewise appears to be a weakness.

**Relation To Broader Scientific Literature:**

Retrieval technology has emerged as a critical advancement across various domains. Consequently, this work on retrieval methods has potential applications spanning multiple scientific areas, underscoring its broader impact beyond the immediate field of study.

**Theoretical Claims:**

The theoretical claims presented in the paper appear sound; however, the work would benefit significantly from more thorough comparison with existing literature.

---

> ### Author Rebuttal · Authors · 2025-04-01
>
> We sincerely appreciate your valuable comments on the advanced applications of retrieval in the NLP domain. However, we believe that the key focus of retrieval tasks differs between the fields of image and NLP. We hope the following response will be helpful to emphasize our contribution to manifold ranking and explain why our experiments focus on image retrieval.
>
> **Q1: Motivation for focusing on image retrieval and its challenges.**
>
> Our proposed method addresses a fundamental problem in manifold ranking, which aims to **utilize the inherent structure of the data space** to improve retrieval performance (as discussed in Section 1, 3). Specifically, manifold ranking assumes that instances within a dataset exhibit intrinsic similarities, and when embedded into a semantic feature space, **similar instances tend to form a low-dimensional manifold**. Unlike the conventional approach that relies solely on pairwise similarity between the query and individual samples, manifold ranking **capitalizes on the latent relationships among instances within the database** to improve retrieval performance.
>
> In image retrieval, each image is represented by a global feature, forming a vector database for retrieval. Refining initial search results by re-extracting image features with a more powerful model is often **computationally expensive and impractical**. Therefore, an efficient strategy is required to enhance retrieval performance by **leveraging the structural relationships among existing feature representations, rather than relying on additional feature extraction.** This characteristic aligns well with the principles of manifold ranking, such that existing algorithms predominantly use it as their primary evaluation benchmark.
>
> Conversely, modern NLP tasks tend to go beyond measuring retrieval effectiveness solely based on textual semantic similarity. For instance, recent benchmarks like "Bright" place greater emphasis on **logical reasoning and deep text comprehension**. These tasks require finer-grained semantic alignment and reasoning ability between words in the query and documents, **which differs from our task setting that relies solely on original global semantic features.** To enhance the quality of retrieval results, current NLP-driven re-ranking methods primarily leverage more advanced models, such as LLMs and Cross Encoders, to **conduct a deeper analysis of the interactions between queries and documents at the token level.** Given that our method is focused on exploring the semantic relationships among candidates based on global features, it is not well-suited for NLP retrieval tasks that necessitate a fine-grained linguistic understanding.
>
> Nevertheless, our approach still remains effective for textual retrieval tasks focused on content search, including text-image retrieval and specific dense model-based information retrieval applications.
>
> **Q2: Broader applications.**
>
> Regarding the concern in Weakness 1 about the applicability of our method to broader retrieval tasks, while it is primarily designed for image retrieval, it is also effective for various textual retrieval tasks with a semantically dense feature database. Leveraging VLMs and pretrained dense retrieval models to process the image and corpus databases, our method can be applied to text-image retrieval and information retrieval tasks. As shown in Tab. 1, 2, our method effectively improves retrieval performance, particularly in text-image retrieval tasks.
>
> Tab 1. Zero-shot text-image retrieval on Flickr30k, based on CLIP and BLIP.
>
> ||R@1|R@5|R@10|
> |:-: |:-:|:-:|:-:|
> |CLIP|55.7|80.8|88.3|
> |CLIP+Ours|66.7|93.6|96.8|
> |BLIP| 68.3 |89.4|94.2|
> |BLIP+Ours|70.4|95.5|98.4|
>
> Tab 2. Information retrieval on ArguAna, based on the pretrained model from BEIR.
>
> ||nDCG@1|nDCG@5|nDCG@10|
> |:-:|:-:|:-:|:-:|
> |Baseline|20.6|37.5|42.6|
> |Ours|27.3|47.0|52.2|
>
> **Q3: Evaluation and datasets.**
>
> ROxf and RPar are among the most widely used datasets in image retrieval, with the hard protocol posing significant challenges. Nearly all retrieval models report their performance on these benchmarks to ensure fair comparisons, and prior reranking studies consistently use R-GeM as a baseline. Additionally, since different retrieval models yield varying levels of feature representations, it is beneficial to assess the generalizability of LPMT.
>
> Evaluating our method on NLP and RAG benchmarks such as Bright and RGB falls beyond the scope of this study. Our approach applies graph theory to model feature databases, ensuring more reliable results by exploiting the underlying manifold information. We believe these benchmarks are more suited for evaluating LLMs and NLP-based rerankers.
>
> **Q4: Literature review.**
>
> We appreciate the recommendation to further discuss NLP-focused studies, which could enhance the overall understanding of the retrieval landscape. We will further discuss their distinctions in the revised version.

---

> > ### Comment · Reviewer_ij2o · 2025-04-03
> >
> > thanks for rebuttal and i have updated my score

---

> > > ### Author Response · Authors · 2025-04-04
> > >
> > > Thanks for your valuable feedback and positive reassessment. We sincerely appreciate your support! Your suggestions will be instrumental in further refining our manuscript.

---

### Official Review · Reviewer_dYaJ · 2025-03-22

**Overall Recommendation:** 4

**Summary:**

In this paper, the authors focused on the diffusion-based re-ranking for instance retrieval. Considering the issue of decaying positive signals and the impact of disconnections in the existing methods, the authors proposed the Locality Preserving Markovian Transition (LPMT) framework. The proposed method consists of three key modules, including BCD, LSE, and TMT. Specifically, BCD integrates diffusion processes across separate graphs to establish strong similarity relationships. LSE encodes each instance into a distribution to enhance local consistency, and TMT connects these distributions through a thermodynamic Markovian transition process for efficient global retrieval while maintaining local effectiveness. Extensive experiments prove the effectiveness of the proposed method.

**Claims And Evidence:**

The authors present a comprehensive set of experiments on multiple datasets with different difficulty levels and using various deep retrieval models. For example, in Table 1, LPMT shows significant improvements in mAP compared to other methods such as AQE, αQE, and CAS under the medium and hard evaluation protocols on ROxf and RPar datasets. The ablation studies also provide evidence for the effectiveness of each component of LPMT.

**Essential References Not Discussed:**

NA

**Experimental Designs Or Analyses:**

The experimental designs are sound. The authors compare LPMT with a wide range of re-ranking methods, including query expansion methods, diffusion-based methods, context-based methods, and learning-based methods.

**Methods And Evaluation Criteria:**

The proposed methods make sense for the problem of instance retrieval. The use of diffusion-based processes in BCD is a well-established approach in the field, and the innovation of integrating multiple graphs through bidirectional collaborative diffusion helps to capture the manifold structure more effectively. LSE's encoding of instances into distributions and TMT's use of a thermodynamic transition process are novel and address the limitations of traditional diffusion-based methods.

**Other Comments Or Suggestions:**

Given the complexity of the method, it would be beneficial if the authors could explore ways to simplify it without sacrificing performance. This could make the method more accessible for practical applications.

**Other Strengths And Weaknesses:**

Strong points:
The paper has a solid theoretical foundation, with detailed mathematical derivations and proofs for the proposed methods.

Weak points:
The proposed method is relatively complex, with multiple components and hyperparameters. This may limit its practical application in some scenarios where computational resources are limited.

**Questions For Authors:**

1. In the BCD component, the optimization process involves iteratively updating the similarity matrix F and weights β. How to ensure the stability of this iterative process?
2. In the TMT component, the use of the Wasserstein distance as an approximation for the minimum transition flow cost is based on certain assumptions. How sensitive is the performance of LPMT to these assumptions?

**Relation To Broader Scientific Literature:**

The key contributions of the paper may add a new perspective to the field of instance retrieval.

**Theoretical Claims:**

The theoretical claims in the paper seem to be well-founded. The authors provide a detailed mathematical derivation for each component of LPMT. It would be beneficial if the authors could provide more intuitive explanations for some of the theoretical concepts, especially for readers who are not familiar with the complex mathematical theories of stochastic thermodynamics.

---

> ### Author Rebuttal · Authors · 2025-04-01
>
> **Q1: Concerns about the involvement of multiple components and hyperparameters.**
>
> Regarding the concern about multiple components and hyperparameters. Although our method consists of multiple modules, each can be implemented in a relatively straightforward manner. For example, the BCD objective can be iteratively solved following Algorithm 1, while TMT reduces to solving an optimization problem (Eq. 20) for the optimal transition strategy. These processes require simple inputs (e.g., BCD only needs the adjacency matrix $W$), ensuring low coupling and easy deployment. Ablation studies demonstrate the robustness of LPMT, showing that most hyperparameters have minimal impact on performance. When adapting to new tasks, only a few key parameters ($k_1$, $k_2$, $\theta$) need adjustment, reducing the tuning effort when adapting to new tasks. Additionally, high-complexity steps can be decoupled and precomputed (see Q2), further improving efficiency in resource-limited scenarios.
>
> **Q2: Simplify the implementation for practical applications.**
>
> Thanks for the concern about the accessibility to practical applications. While our primary focus in this work is to improve the effectiveness of manifold ranking, there are some feasible optimizations to simplify the implementation or enhance efficiency.
>
> Firstly, BCD module can be replaced with Gaussian kernel function to approximate the manifold-aware similarity when computation resource is limited. Additionally, we can adopt an offline strategy to precompute high complexity operations within the database, such that each image can be embed with the diffusion-based similarity in advance. When a new query arrives, we can approximate its diffusion-based similarity with other instances by linearly aggregating neighboring samples. Similarly, the TMT distance can also be precomputed within the graph, allowing us to perform online reranking with an efficiency close to kNN. We will continue to investigate methods to simplify the deployment of the reranking system without sacrificing performance.
>
> **Q3: How to ensure the convergence when jointly optimize $\beta$ and $F$ in BCD.**
>
> The convergence of the optimization is ensured by the following factors:
>
> (1) When $\beta$ is fixed, the objective function (denoted as $J(\beta, F) = \lambda\|\beta\|_2^2/2 + \sum\beta _ {v}H^{v}$) is convex with respect to $F$ and admits a unique minimum point. Similarly, when $F$ is fixed, the function is convex with respect to $\beta$ and also has a unique minimum.
>
> (2) In our optimization, we construct a sequence of $\beta^{(k)}$ and $F^{(k)}$ using the following update rules:
> $$
> \beta^{(k)} = \arg\min\limits_{\beta} J(\beta, F^{(k)}),$$
> $$
> F^{(k+1)} = \arg\min\limits_{F} J(\beta^{(k)}, F).$$
> The resulting variable sequence is $\{(\beta^{(0)}, F^{(0)}), ..., (\beta^{(k)}, F^{(k)}), (\beta^{(k)}, F^{(k+1)}), (\beta^{(k+1)}, F^{(k+1)}), ...\}$. The corresponding sequence of objective values $J$ is non-increasing.
>
> (3) If $J(\beta^{(k)}, F^{(k)}) = J(\beta^{(k)}, F^{(k+1)})$, then by the uniqueness of the optimal point discussed in (1), we have $F^{(k)} = F^{(k+1)}$, and subsequent updates of $F$ and $\beta$ will remain unchanged. The same holds if $J(\beta^{(k)}, F^{(k+1)}) = J(\beta^{(k+1)}, F^{(k+1)})$.
>
> (4) If the condition in (3) never occurs, then the sequence of objective values strictly decreases at each iteration. Since $J \geq 0$, the sequence converges to a finite value $J^*$.
>
> **Q4: Assumptions for TMT component and their influence to the performance.**
>
> In our proposed TMT component, we assume that each transition is governed by a master equation and occurs within a local region. As demonstrated in Appendix B, we prove that the transition cost between two neighboring distributions is equivalent to the Wasserstein distance, where the distance on the graph serves as the cost matrix. This locality assumption mitigates information loss over multiple iterations of the diffusion process and effectively models long-term transitions between distributions in the graph. Compared to directly computing the flow cost between distant distributions, our approach ensures the reliability of each transition while reducing the computational complexity over the entire graph, thereby improving both performance and efficiency.
>
> **Q5: Suggestions for providing more intuitive explanations and improving the organization of supplementary materials and image labels.**
>
> Thank you very much for the valuable feedback. We appreciate your suggestions and will incorporate more intuitive explanations of the theoretical concepts to enhance clarity. In the revised version, we will also reorganize the image captions and supplementary materials for better readability and coherence.

---

> > ### Comment · Reviewer_dYaJ · 2025-04-07
> >
> > Thanks for your detailed response. I have updated the score.

---

> > > ### Author Response · Authors · 2025-04-07
> > >
> > > Thanks very much for your kind and constructive feedback. We sincerely appreciate the time and effort you dedicated to reviewing our paper. We will carefully revise the manuscript based on your suggestions.

---

### Decision · Program_Chairs · 2025-05-01

**Decision:**

Accept (poster)

**Comment:**

This paper introduces the Locality Preserving Markovian Transition (LPMT) framework, employing a long-term thermodynamic transition process with multiple states for accurate manifold distance measurement. After the rebuttal, it received two accept and two weak accept. Its merits, including the clear motivation, solid theoretical foundation, and good experimental results, are well recognized by the reviewers. The response well addresses the reviewers' concerns about the convergence, broader application, and so on. I think the current manuscript meets the requirement of this top conference and recommend for acceptance. Please incorporate the revision in the updated manuscript.